# FAN: Fourier Analysis Networks

**Yihong Dong**[1][*], **Ge Li**[1][*], **Yongding Tao**[1], **Xue Jiang**[1], **Kechi Zhang**[1], **Jia Li** ♂[1],
**Jinliang Deng**[2], **Jing Su**[3], **Jun Zhang**[3], **Jingjing Xu**[3]
[1]School of Computer Science, Peking University
[2]The Hong Kong University of Science and Technology    [3]ByteDance
dongyh@stu.pku.edu.cn, lige@pku.edu.cn

## Abstract

Despite the remarkable successes of general-purpose neural networks, such as MLPs and Transformers, we find that they exhibit notable shortcomings in modeling and reasoning about periodic phenomena, achieving only marginal performance within the training domain and failing to generalize effectively to out-of-domain (OOD) scenarios. Periodicity is ubiquitous throughout nature and science. Therefore, neural networks should be equipped with the essential ability to model and handle periodicity. In this work, we propose FAN, a novel neural network that effectively addresses periodicity modeling challenges while offering broad applicability similar to MLP with fewer parameters and FLOPs. Periodicity is naturally integrated into FAN's structure and computational processes by introducing the Fourier Principle. Unlike existing Fourier-based networks, which possess particular periodicity modeling abilities but face challenges in scaling to deeper networks and are typically designed for specific tasks, our approach overcomes this challenge to enable scaling to large-scale models and maintains the capability to be applied to more types of tasks. Through extensive experiments, we demonstrate the superiority of FAN in periodicity modeling tasks and the effectiveness and generalizability of FAN across a range of real-world tasks. Moreover, we reveal that compared to existing Fourier-based networks, FAN accommodates both periodicity modeling and general-purpose modeling well.

## 1 Introduction

The flourishing of modern machine learning and artificial intelligence is inextricably linked to the revolutionary advancements in the foundational architecture of general-purpose neural networks. For instance, multi-layer perceptron (MLP) [Rosenblatt, 1958, Haykin, 1998] plays a pivotal role in laying the groundwork for current deep learning models, with its expressive power guaranteed by the universal approximation theorem [Hornik et al., 1989]. Recent claims about the impressive performance of large models on various tasks are typically supported by Transformer architecture [Vaswani et al., 2017, Touvron et al., 2023, OpenAI, 2023]. In this context, the community's enthusiasm for research on neural networks has never diminished. Some emerged neural networks demonstrate notable capabilities in specific fields [Gu and Dao, 2023, Liu et al., 2024], sparking widespread discussion within the community.

Beneath the surface of apparent prosperity, we uncover a critical issue that remains in existing general-purpose neural networks: *they struggle to model the periodicity from data, especially in OOD*

---

[*]Equal Contribution

[†]This work was supported by a cooperation project between Peking University and ByteDance Company. During this time, Yihong was also an intern at ByteDance.

[‡]The code is available at `https://github.com/YihongDong/FAN`

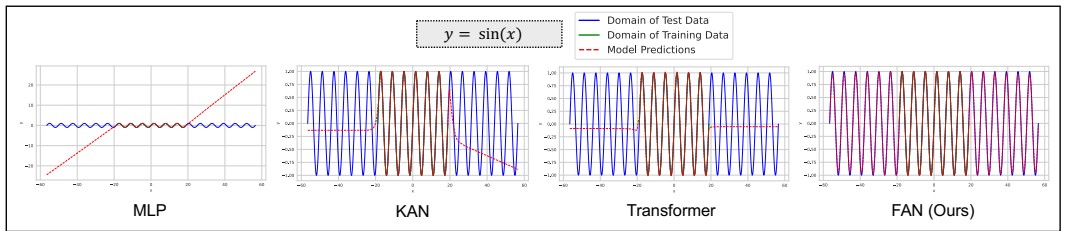

Figure 1: The performance of different neural networks within and outside the domain of their training data for the sine function, where $x$ is a scalar variable.

*scenarios*. We showcase this issue through an empirical study as illustrated in Figure 1. The results indicate that existing neural networks, including MLP [Rosenblatt, 1958], KAN [Liu et al., 2024], and Transformer [Vaswani et al., 2017], face difficulties in fitting periodic functions, even on a simple sine function. Although they demonstrate some proficiency in interpolation within the domain of training data, they tend to falter when faced with extrapolation challenges of test data. This signifies that their generalization capacity is primarily dictated by the scale and diversity of the training data, rather than by the learned principles of periodicity to perform reasoning.

Periodicity is an essential characteristic in various forms of reasoning and generalization, as it provides a basis for predictability in many natural and engineered systems by leveraging recurring patterns in observations. Besides periodic phenomena, non-periodic phenomena can also be contextualized or explained within some larger or more macro-periodic framework. Although some Fourier-based networks exhibit particular periodic modeling abilities, they are primarily tailored for specific tasks [Silvescu, 1999, Liu, 2013] and do not work well as the networks deepen [Liu et al., 2020], which limits their applicability to the general task such as language modeling [Uteuliyeva et al., 2020]. However, our goal is to exploit periodicity to benefit a broader range of tasks including language modeling. To achieve this, *we aim to develop a neural network that accommodates modeling and reasoning capabilities for periodicity while maintaining the capability to be applied to more types of tasks.*

In this paper, we propose Fourier Analysis Network (FAN), a novel neural network built upon the principle of Fourier Analysis. By leveraging the power of Fourier Series, we enable the neural network to model periodic patterns and extrapolate beyond them, offering the network a way to model the general principles from the data. FAN follows two core principles, the first ensures that its periodic modeling capacity scales with network depth, while the second guarantees periodic modeling is available throughout the network. These principles allow it to scale to deeper networks, a capability where existing Fourier neural networks fall short. As a result, FAN exhibits exceptional capabilities in periodicity modeling, while maintaining broad applicability to the general task, which holds great potential as a substitute for MLP, with fewer parameters and FLOPs.

To verify the effectiveness of FAN, we conduct extensive experiments from three main aspects: 1) For periodicity modeling, FAN achieves significant improvements in fitting both basic and complex periodic functions, compared to existing neural networks (including MLP, KAN, and Transformer), particularly in OOD scenarios. 2) FAN shows superior performance in various real-world tasks, such as symbolic formula representation, time series forecasting, and language modeling. Using FAN outperforms the representative models in various tasks, including MLP, KAN, LSTM, Mamba, and Transformer. 3) Compared to existing Fourier-based networks, FAN accommodates both periodicity modeling and general-purpose modeling well. The advantageous characteristics and promising results indicate that FAN has the potential to become a basic component for building fundamental large models.

## 2 Preliminary Knowledge

Fourier Analysis [Stein and Weiss, 1971, Duoandikoetxea, 2024] is a mathematical framework that decomposes functions into their constituent frequencies, revealing the underlying periodic structures within complex functions. At the heart of this analysis lies Fourier Series [Tolstov, 2012], which expresses a periodic function as an infinite sum of sine and cosine terms. Mathematically, for a

function $f(x)$, its Fourier Series expansion can be represented as:

$$f(x) = a_0 + \sum_{n=1}^{\infty} \left( a_n \cos\left(\frac{2\pi n x}{T}\right) + b_n \sin\left(\frac{2\pi n x}{T}\right) \right), \tag{1}$$

where $T$ is the period of the function, and the coefficients $a_n$ and $b_n$ are determined by integrating the function over one period:

$$a_n = \frac{1}{T} \int_0^T f(x) \cos\left(\frac{2\pi n x}{T}\right) dx, \quad b_n = \frac{1}{T} \int_0^T f(x) \sin\left(\frac{2\pi n x}{T}\right) dx. \tag{2}$$

The power of Fourier Series lies in its ability to represent a wide variety of functions, including non-periodic functions through periodic extensions, enabling the extraction of frequency components. Building on this math foundation, FAN aims to embed the periodic characteristics into network architecture, enhancing generalization capabilities and performance on various tasks, particularly in scenarios requiring the identification of patterns and regularities.

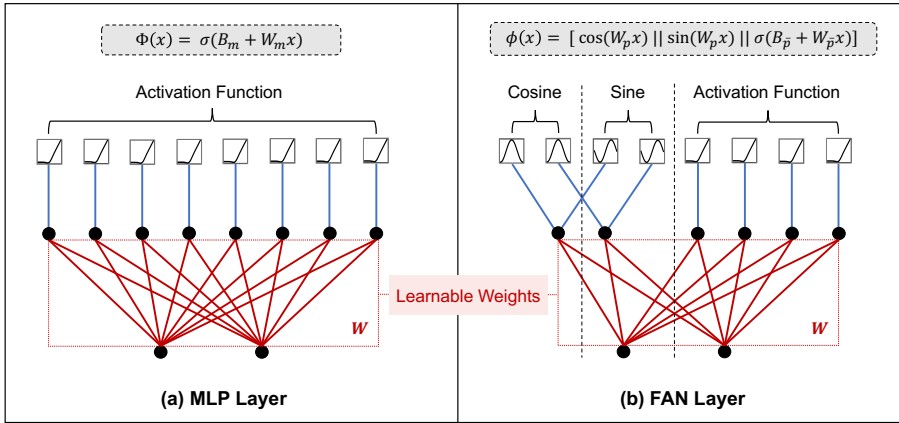

Figure 2: Illustrations of FAN layer $\phi(x)$ vs. MLP layer $\Phi(x)$.

## 3 Fourier Analysis Network (FAN)

In this section, we first construct a naive neural network modeled by the formula of Fourier Series. Then, by modifying and improving it, we design FAN adhering to two core principles. Finally, we discuss the difference between the FAN layer and MLP layer.

Consider a task involving input-output pairs $\{x_i, y_i\}$, with the objective of identifying a function $f(x) : \mathbb{R}^{d_x} \to \mathbb{R}^{d_y}$ that approximates the relationship such that $y_i \approx f(x_i)$ for all $x_i$, where $d_x$ and $d_y$ denote the dimensions of $x$ and $y$, respectively. We first construct a shallow neural network $f_S(x)$ that represents Fourier Series expansion of the function, specifically $\mathcal{F}\{f(x)\}$, as described in Eq. (1), we can express $f_S(x)$ as follows:

$$
\begin{aligned}
f_S(x) &\triangleq a_0 + \sum_{n=1}^{N} \left( a_n \cos\left(\frac{2\pi n x}{T}\right) + b_n \sin\left(\frac{2\pi n x}{T}\right) \right), \\
&\overset{(I)}{=} a_0 + \sum_{n=1}^{N} \left( w_n^c \cos\left(w_n^{\text{in}} x\right) + w_n^s \sin\left(w_n^{\text{in}} x\right) \right), \\
&\overset{(II)}{=} B + [w_1^c, w_2^c, \cdots, w_n^c] \cos([w_1^{\text{in}}||w_2^{\text{in}}|| \cdots ||w_n^{\text{in}}]x) \\
&\quad + [w_1^s, w_2^s, \cdots, w_n^s] \sin([w_1^{\text{in}}||w_2^{\text{in}}|| \cdots ||w_n^{\text{in}}]x) \\
&= B + W_c \cos(W_{\text{in}} x) + W_s \sin(W_{\text{in}} x), \\
&\overset{(III)}{=} B + W_{\text{out}} [\cos(W_{\text{in}} x) || \sin(W_{\text{in}} x)],
\end{aligned}
\tag{3}
$$

Table 1: Comparison of FAN layer and MLP layer, where $d_{\mathrm{p}}$ is a hyperparameter of FAN layer and defaults to $\frac{1}{4}d_{\text{output}}$ in this paper, $d_{\text{input}}$ and $d_{\text{output}}$ denote the input and output dimensions of the neural network layer, respectively. In our evaluation, the floating point of operations (FLOPs) for any arithmetic operations are considered as 1, and for Boolean operations as 0.

| | MLP Layer | FAN layer |
|---|---|---|
| Formula | $\Phi(x) = \sigma(B_m + W_m x)$ | $\phi(x) = [\cos(W_p x)\|\|\sin(W_p x)\|\|\sigma(B_{\bar{p}} + W_{\bar{p}}x)]$ |
| Num of Params | $(d_{\text{input}} \times d_{\text{output}}) + d_{\text{output}}$ | $(1 - \frac{d_p}{d_{\text{output}}}) \times ((d_{\text{input}} \times d_{\text{output}}) + d_{\text{output}})$ |
| FLOPs | $2 \times (d_{\text{input}} \times d_{\text{output}})$ $+\text{FLOPS}_{\text{non-linear}} \times d_{\text{output}}$ | $(1 - \frac{d_p}{d_{\text{output}}}) \times 2 \times (d_{\text{input}} \times d_{\text{output}})$ $+\text{FLOPS}_{\text{non-linear}} \times d_{\text{output}}$ |

where $B \in \mathbb{R}^{d_y}, W_{\text{in}} \in \mathbb{R}^{N \times d_x}$, and $W_{\text{out}} \in \mathbb{R}^{d_y \times 2N}$ are learnable parameters, (I) follows that the computation of $a_n$ and $b_n$ computed via Eq. (2) is definite integral, (II) and (III) follows the equivalence of the matrix operations, $[\cdot\|\|\cdot]$ and $[\cdot, \cdot]$ denotes the concatenation along the first and second dimension, respectively.

To fully leverage the advantages of deep learning, we can stack the aforementioned network $f_{\mathrm{S}}(x)$ to form a deep network $f_{\mathrm{D}}(x)$, where the $i$-th layer, denoted as $l_i(x)$, retains the same structural design as $f_{\mathrm{S}}(x)$. Therefore, $f_{\mathrm{D}}(x)$ can be formulated as:

$$f_{\mathrm{D}}(x) = l_L \circ l_{L-1} \circ \cdots \circ l_1 \circ x, \tag{4}$$

where $l_1 \circ x$ denotes the application of the left function $l_1$ to the right input $x$, that is $l_1(x)$. However, we discover that the direct stacking of $f_{\mathrm{S}}(x)$ results in the primary parameters of the network $f_{\mathrm{D}}(x)$ focusing on learning the angular frequency ($\omega_n = \frac{2\pi n}{T}$), thereby neglecting the learning of the Fourier coefficients ($a_n$ and $b_n$), as follows:

$$\begin{aligned} f_{\mathrm{D}}(x) &= l_L(l_{L-1} \circ l_{L-2} \circ \cdots \circ l_1 \circ x) \\ &= B^L + W_{\text{out}}^L[\cos(W_{\text{in}}^L(l_{1:L} \circ x)\|\|\sin(W_{\text{in}}^L(l_{1:L} \circ x))] \end{aligned} \tag{5}$$

where $l_{1:L} \circ x$ is defined as $l_{L-1} \circ l_{L-2} \circ \cdots \circ l_1 \circ x$, $W_{\text{in}}^L(l_{1:L} \circ x)$ is used to approximate the angular frequencies, and $W_{\text{out}}^L$ is used to approximate the Fourier coefficients. We can find that the capacity of $f_{\mathrm{D}}(x)$ to fit the Fourier coefficients is independent of the depth of $f_{\mathrm{D}}(x)$, which is an undesirable outcome. It will limit the network's representation ability, hindering to address the complex tasks.

To this end, we design FAN based on the following principles: 1) the capacity of FAN to represent the Fourier coefficients should be positively correlated to its depth; 2) the output of any hidden layer can be employed to model periodicity using Fourier Series through the subsequent layers. The first one enhances the expressive power of FAN for periodicity modeling by leveraging its depth, while the second one ensures that the features of FAN's intermediate layers are available to perform periodicity modeling.

Suppose we decouple $f_{\mathrm{S}}(x)$ as follows:

$$f_{\mathrm{S}}(x) = f_{out} \circ f_{in} \circ x, \tag{6}$$

where

$$f_{in}(x) = [\cos(W_{\text{in}}x)\|\|\sin(W_{\text{in}}x)], \tag{7}$$
$$f_{out}(x) = B + W_{\text{out}}x. \tag{8}$$

To satisfy both principles, the inputs of the intermediate layers in FAN necessitate to employ $f_{in}$ and $f_{out}$ simultaneously, rather than applying them sequentially.

Finally, FAN is designed on this basis, with the FAN layer $\phi(x)$ defined as below:

$$\phi(x) \triangleq [\cos(W_p x)\|\|\sin(W_p x)\|\|\sigma(B_{\bar{p}} + W_{\bar{p}}x)], \tag{9}$$

where $W_p \in \mathbb{R}^{d_x \times d_p}, W_{\bar{p}} \in \mathbb{R}^{d_x \times d_{\bar{p}}}$, and $B_{\bar{p}} \in \mathbb{R}^{d_{\bar{p}}}$ are learnable parameters (with the hyperparameters $d_p$ and $d_{\bar{p}}$ indicating the first dimension of $W_p$ and $W_{\bar{p}}$, respectively), the layer output $\phi(x) \in \mathbb{R}^{2d_p + d_{\bar{p}}}$, and $\sigma$ denotes the activation function. Under this definition, the MLP layer can be

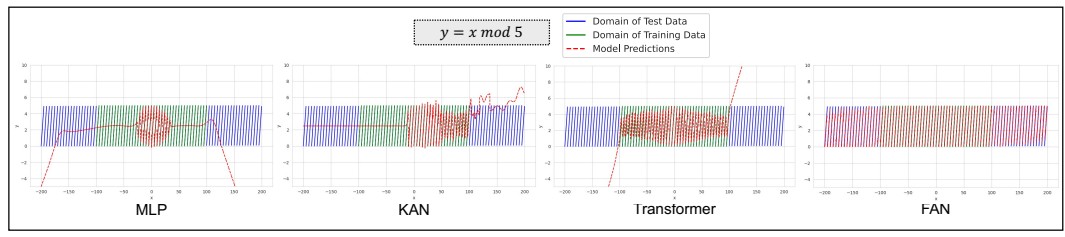

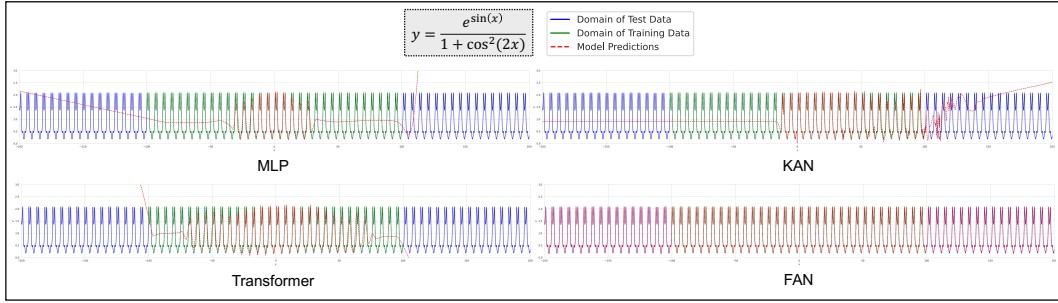

Figure 3: The performance of FAN in periodicity modeling compared to MLP, KAN, and Transformer (Part I), where the green line represents the test data within the domain of training data, while the blue line represents the test data outside the domain of training data.

regarded as a special form of Eq. (9), when $W_p$ are learned to be zero metrics, which provides a way for FAN to maintain general-purpose modeling abilities as MLP.

The entire FAN is defined as the stacking of the FAN layer $\phi(x)$ as follows:

$$\text{FAN}(x) = \phi_L \circ \phi_{L-1} \circ \cdots \circ \phi_1 \circ x, \tag{10}$$

where

$$\phi_l(x) = \begin{cases} [\cos(W_p^l x) || \sin(W_p^l x) || \sigma(B_{\bar{p}}^l + W_{\bar{p}}^l x)], & \text{if } l < L, \\ B^L + W^L x, & \text{if } l = L, \end{cases} \tag{11}$$

**The difference between FAN and MLP.** The illustrations of FAN layer $\phi(x)$ vs. MLP layer $\Phi(x)$ are shown in Figure 2. Note that the FAN layer $\phi(x)$ computed via Eq. (9) can seamlessly replace the MLP layer $\Phi(x)$ computed via Eq. (12) in various models with fewer parameters and FLOPs, achieved by sharing the parameters and computation of Sin and Cos parts. The number of parameters and FLOPs of the FAN layer compared to the MLP layer are presented in Table 1. The reduction ratio of parameters and FLOPs is about $\frac{d_p}{d_{\text{output}}}$, which is set to $\frac{1}{4}$ by default in this paper.

## 4 Experiments

In this section, we first verify the superiority of FAN in periodicity modeling tasks (Section 4.1). Second, we demonstrate the effectiveness and generalizability of FAN across a range of real-world tasks (Section 4.2). Finally, we conduct further analysis of FAN (Section 4.3), including comparisons with Fourier-based networks, running time, hyperparameter impact, and more. See Appendix B for more experiments and the experimental details can be found in Appendix C.

### 4.1 Periodicity Modeling

**Setup.** In periodic modeling tasks, we select periodic functions with practical significance and compare the models' performance in learning the underlying principles of periodicity. Specifically, we generate data from periodic functions over a large domain, using a portion of this domain as training data and the entire domain as test data, i.e., a part of test data would be out of the domain of

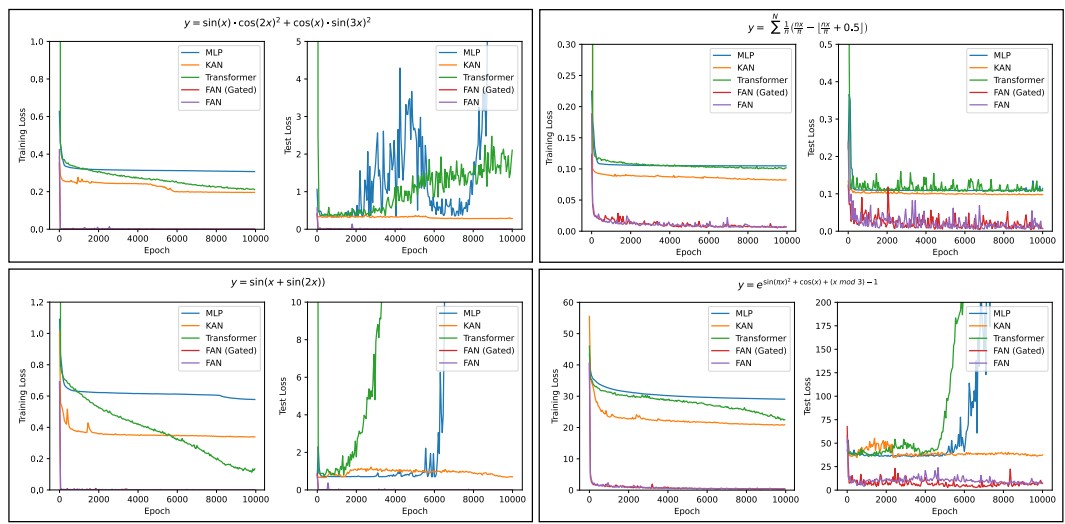

Figure 4: Comparison of training and test losses for different models on the tasks of learning complex periodic functions.

training data. We compare FAN and its variant FAN (Gated)[§], with MLP, KAN, and Transformer. The input of this task is scalar.

**Results.** Figure 3, as well as Figure 6 in Appendix, show the performance of FAN and other baselines in periodicity modeling. The results indicate that existing neural networks, including MLP, KAN, and Transformers, exhibit notable deficiencies in their ability to model periodicity. Although they attempt to fit these periodic functions, their ability limits their performance in modeling a large domain of periodicity, including the test data within and outside the domain of the training data. In contrast, FAN significantly outperforms baselines in all these tasks of periodicity modeling. Moreover, FAN performs exceptionally well on the test data both within and outside the domain, indicating that our specialized design of FAN can effectively model and understand periodicity rather than merely memorize the training data.

We also compare the training process of different models on the tasks of learning complex periodic functions, as shown in Figure 4, which leads to the following findings. 1) FAN far exceeds the other baselines in both convergence speed and final effects. 2) FAN (Gated) often achieves faster convergence than FAN, but the final performance remains comparable. 3) Although the baselines show stabilization or gradual reductions in training loss as the number of epochs increases, their modeling may have diverged considerably from the distribution of the test data, resulting in a sharp increase in test loss. This phenomenon further demonstrates the shortcomings of these models in capturing periodicity.

## 4.2 Application of Real-world Task

**1) Symbolic Formula Representation** is a common task in both mathematics and physics. We follow the experiments conducted in KAN's paper [Liu et al., 2024], adhering to the same tasks, data, hyperparameters, and baselines. In addition to the original baselines, we also include Transformer for comparison in this task.

**Results.** Figure 7 in Appendix shows the performance of different models applied to common functions in mathematics and physics. We can observe that while KAN remains competitive with FAN when the number of parameters is small, its performance declines clearly as the number of parameters increases, which exhibits a U-shaped trend [Liu et al., 2024]. In contrast, as the number of parameters becomes large, FAN consistently outperforms the other baselines, including MLP, KAN, and Transformer, in fitting these functions, despite many of these functions being only partially periodic or even implicitly periodic. This may be attributed to FAN's ability to capture and model both

---

[§]FAN (Gated) is a variant of FAN that adds gates to control the tendency of the layer, with the formula defined as $\phi_g(x) = [g \cdot \cos(W_p x) || g \cdot \sin(W_p x) || (1 - g) \cdot \sigma(B_{\bar{p}} + W_{\bar{p}} x)]$, where $g$ is a learnable parameter.

periodic and non-periodic features and the advantages of fewer parameters. These results indicate that although FAN enhances its ability to model periodicity, it does not compromise its capacity to fit non-periodic functions.

**2) Time Series Forecasting** plays a critical role in various real-world applications. We employ four public datasets of this task to assess the model performance on time series forecasting, including Weather [Wu et al., 2021], Exchange [Lai et al., 2018], Traffic [Wu et al., 2021], and ETTh [Zhou et al., 2021] datasets. For each dataset, we input 96 previous time steps and forecast the subsequent time steps of {96, 192, 336, 720}. In this task, we choose the sequence models as baselines, including LSTM, Mamba, and Transformer.

**Results.** As shown in Table 2 (See Table 6 in Appendix for complete results), we compare the performance of Transformer with FAN and other baselines for time series forecasting tasks. The results indicate that Transformer with FAN outperforms other representative sequence models in these tasks. The improvements of Transformer with FAN and FAN (Gated) over the standard Transformer are notable, with the average relative improvements ranging from 15.0% to 15.6% for MSE and from 7.6% to 8.4% for MAE. It suggests that incorporating explicit periodic pattern encoding within neural networks improves time series forecasting performance in real-world applications.

Table 2: Average performance on different public datasets and output lengths in time series forecasting tasks, where Input Length = 96 and the **bold** value indicates the best performance.

| Model | Num of Params | Average | |
| --- | --- | --- | --- |
| | | MSE ↓ | MAE ↓ |
| LSTM | 12.51M | 1.083 | 0.726 |
| Mamba | 12.69M | 1.002 | 0.668 |
| Transformer | 12.12M | 0.994 | 0.689 |
| w/ FAN (Gated) | 11.07M | 0.845 | 0.637 |
| w/ FAN | **11.06M** | **0.839** | **0.631** |
| Improvements | ↓ 1.06M | ↓ 15.6% | ↓ 8.4% |

**3) Language Modeling** is a fundamental task in natural language processing. We conduct language modeling using the SST-2 [Socher et al., 2013] dataset and evaluate the model's performance on its test set, as well as on the related datasets such as IMDB [Maas et al., 2011], Sentiment140 [Sahni et al., 2017], and Amazon Reviews [Linden et al., 2003]. These four classic datasets all belong to the field of sentiment analysis. The comparisons are between Transformer with FAN and FAN (Gated), along with the classic sequence models, including LSTM, Mamba, and Transformer.

Table 3: Performance of different sequence models on language modeling tasks, where the models are trained on the training set of SST-2 and evaluated on the other datasets, the **bold** value indicates the best performance on each column, the ***bold italic*** indicates the second-best performance, and the improvements represent relative improvements of using FAN based on standard Transformer.

| Model | Num of Params | SST-2 (test) | | IMDB | | Sentiment140 | | Amazon Reviews | |
| --- | --- | --- | --- | --- | --- | --- | --- | --- | --- |
| | | Loss ↓ | Acc ↑ | Loss ↓ | Acc ↑ | Loss ↓ | Acc ↑ | Loss ↓ | Acc ↑ |
| LSTM | 120.14M | 0.4760 | 80.60 | 0.6449 | 64.38 | 0.8026 | 59.79 | 0.5791 | 71.52 |
| Mamba | 129.73M | 0.4335 | 79.59 | 0.6863 | 62.03 | **0.7871** | 58.74 | 0.6163 | 67.19 |
| Transformer | 109.48M | *0.4297* | *81.19* | *0.5649* | 69.94 | 0.8891 | 57.79 | 0.5563 | 71.55 |
| w/ FAN (Gated) | 95.33M | 0.4250 | 80.39 | 0.5817 | *70.12* | *0.7941* | **61.94** | *0.4835* | *76.89* |
| w/ FAN | **95.32M** | **0.4094** | **81.54** | **0.5225** | **73.98** | 0.8257 | *60.93* | **0.4748** | **77.63** |
| Improvements | ↓ 14.16M | ↓ 4.72% | ↑ 0.43% | ↓ 7.51% | ↑ 5.78% | ↓ 7.13% | ↑ 5.43% | ↓ 14.65% | ↑ 8.50% |

**Results.** We report the performance comparison between different sequence models across four sentiment analysis datasets, as shown in Table 3. The results indicate that Transformer with FAN achieves clear improvements compared to the standard Transformer and other baselines, such as LSTM and Mamba, especially for zero-shot OOD performance on IMDB, Sentiment140, and Amazon Reviewers datasets. Using FAN achieves the relative improvements up to 14.65% and 8.50% in terms of Loss and Accuracy respectively, while reducing parameter numbers by about 14.16M. It indicates the potential of periodicity modeling to enhance both effectiveness and generalization on cross-domain language modeling and sentiment analysis tasks.

## 4.3 Further Analysis of FAN

**Comparison with Fourier-based Networks.** We compare FAN with Fourier-based networks in terms of their periodicity modeling abilities and general-purpose capabilities for language modeling. Some previous works have explored the application of Fourier-based Networks in specific tasks [Oreshkin et al., 2020, Tancik et al., 2020, Sitzmann et al., 2020, Han et al., 2022], but these studies primarily involved shallow/small-scale models (i.e., fewer than 1M parameters). Assessing their general modeling capabilities requires evaluating their effectiveness in deeper/larger architectures, we categorize these Fourier-based networks into three main types and systematically evaluate them within the 12-layer Transformer. Specifically, we compare with: 1) **Fourier Neural Network (FNN)** [Silvescu, 1999] using the cosine or sine function or their linear combinations as the activation function, such as SIREN [Sitzmann et al., 2020]. 2) **Fourier Series Neural Network (FSNN)** is defined as Eq. (3), which shares the parameters and computation of sine and cosine part. 3) **Fourier Transform Neural Network (FTNN)** is a type of neural network that employs Fourier Transform to process the intermediate output in the neural network, such as FNO [Li et al., 2021].

Figure 5: Comparison FAN with Fourier-based Networks on complex periodicity modeling ($y = e^{\sin(\pi x)^2 + \cos(x) + (x \mod 3) - 1}$) and language modeling.

Table 4: Comparison FAN with Fourier-based Networks on language modeling tasks, where each of them replaces the MLP layer in the standard transformer and ID means in-domain.

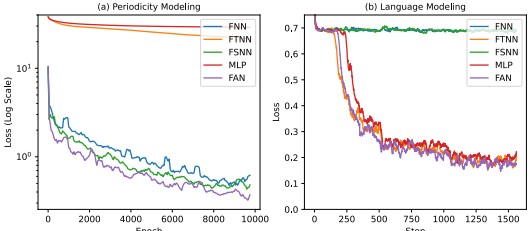

| Model | Num of Params | Loss ↓ | | |
|---|---|---|---|---|
| | | Train | ID Test | OOD Test |
| MLP | 109.48M | 0.2574 | 0.4297 | 0.5649 |
| FNN | 109.48M | 0.6933 | 0.7103 | 0.7135 |
| FSNN | **95.32M** | 0.6931 | 0.7210 | 0.7249 |
| FTNN | 300.56M | 0.2449 | 0.4547 | 0.8128 |
| FAN | **95.32M** | **0.2434** | **0.4094** | **0.5225** |

As shown in Figure 5, only FAN achieves excellent performance on both tasks, indicating the superiority of our specially designed architecture of FAN. In contrast, FNN and FSNN cannot fit language modeling tasks, which aligns with previous work [Uteuliyeva et al., 2020, Liu et al., 2020] and our findings derived from Eq. (3)-(5). Moreover, FTNN performs poorly on complex periodic modeling tasks, akin to MLP. This may be attributed to the fact that FTNN does not incorporate the Fourier principle into the network but applies Fourier Transform as an intermediate processing step, which disadvantages FTNN in capturing periodicity. From Table 4, FAN also achieves fewer parameters and better performance than FTNN in language modeling tasks.

Table 5: Comparison of actual runtime between FAN and MLP under different input/output dimensions (e.g., 8192×8192 indicates both input and output dimensions are 8192).

| | 1024×1024 | 2048×2048 | 4096×4096 | 8192×8192 |
|---|---|---|---|---|
| MLP | **0.064 ms** | **0.114 ms** | 0.212 ms | 0.938 ms |
| FAN | 0.128 ms | 0.133 ms | **0.211 ms** | **0.704 ms** |

**Runtime of FAN.** We analyze the actual running time of FAN layer compared to MLP Layer with different input and output dimensions, as shown in Table 5. The experimental results show that MLPs exhibit smaller runtimes when the input and output sizes are small, due to PyTorch's optimization of MLP. However, as the input and output sizes continue to increase, matrix computations become the main contributor to runtime. At this point, FAN's fewer parameters and reduced FLOPs begin to show significant advantages. Note that FAN can be further optimized from the underlying implementation.

**The impact of hyperparameter $d_p$.** In our experiments, we fix $d_p = \frac{1}{4}d_h$ intuitively for FAN, where $d_h$ denotes the dimension of hidden layers. As shown in Figure 8 of Appendix, we investigate the impact of varying $d_p$ empirically on task performance by changing itself. The results indicate that performance initially improves as $d_p$ increases, but then decreases beyond a certain point. This trend may be attributed to the number of potential periodic features specific to each task. Furthermore, there remains room for further improvements with the better setup of $d_p$.

## 5 Related Work

In this section, we outline the two most relevant directions and associated papers of this work.

**Learning Periodicity with Neural Networks.** Periodic functions are one of the most basic functions of importance to human society and natural science [Newton, 1687, Osborn and Sensier, 2002, Kwasnicki, 2008, De Groot and Franses, 2012, Zhang et al., 2017]. However, commonly used neural networks, such as MLPs and transformers, struggle with modeling periodicity. This limitation is attributed to the lack of inherent "periodicity" in their inductive biases. Some previous works [Silvescu, 1999, Liu, 2013, Parascandolo et al., 2016, Uteuliyeva et al., 2020] proposed merely using standard periodic functions themselves or their linear combinations as activation functions, which only work well on some shallow and simple models. On this basis, work [Liu et al., 2020] introduced the Snake function, i.e., $x + \sin^2(x)$, as the activation function. However, it can fit periodic functions to a certain extent, but its effect is limited especially for OOD scenarios, as demonstrated in Appendix D. Therefore, although some previous studies have attempted to integrate periodic information into neural networks, their actual performance and range of applications remain heavily constrained.

**Fourier-based Neural Network.** Previous studies have explored Fourier-based networks, but these networks generally perform well on specific tasks, while their performance on more general tasks tends to be poorer [Zuo and Cai, 2005, Tan, 2006, Uteuliyeva et al., 2020, Jiang et al., 2022, Chen et al., 2022]. Fourier Neural Networks employ the cosine [Silvescu, 1999, Ngom and Marin, 2021] or sine function [Parascandolo et al., 2016, Sitzmann et al., 2020] or their combination [Liu, 2013] as the activation function. Some work employs Fourier Transform to process the intermediate output of network [Li et al., 2021, Lee-Thorp et al., 2022], but they did not address the challenges of periodicity modeling. Some researches focus on leveraging the network to simulate the formula of Fourier Series [Rafajłowicz and Pawlak, 1997, Halawa, 2008, Lee et al., 2021], which generally possesses a similar principle as Eq. (3). However, this leads to the same problem as in Eq. (5), i.e., they are hard to serve as building blocks for deep neural networks, which limits these approaches' capabilities. More detailed discussion can be found in Appendix G.

In this paper, we design FAN to address these challenges, which performs exceptionally well on periodicity modeling and maintains broad applicability on real-world tasks.

## 6 Discussion

In this section, we have a broad discussion on expressive power, extrapolation capability, and application scope of FAN as follows: ❶ FAN theoretically possesses the equal expressive power as MLP since it also adheres to Universal Approximation Theorem, which guarantees its capacity for functional approximation (refer to Appendix E for the detailed explanation). Moreover, FAN introduces an important enhancement by incorporating periodicity, a feature absent in MLPs. By leveraging this special design, FAN not only retains the capabilities of MLP but also enhances its ability to capture periodic characteristics in data. ❷ We observe that existing networks often exhibit divergent predictions in OOD scenarios, as shown in Figures 3, 4, and 6 for periodicity modeling tasks. In contrast, FAN demonstrates strong OOD extrapolation ability in both periodicity modeling and some real-world tasks. This extrapolation ability indicates that the network is no longer restricted to the paradigms present in training dataset, but instead exhibits a kind of "transboundary thinking". This could be an important avenue for improving generalization and learning efficiency. ❸ Beyond tasks that explicitly require periodicity modeling, FAN also has utility in a broader range of applications, which has been evidenced by our extensive experiments on real-world tasks, such as symbolic formula representation, time series forecasting, language modeling, and image recognition, where FAN achieve competitive or superior performance than Transformers and other baselines. In fact, many machine learning tasks may harbor hidden forms of periodicity, even without explicitly including periodicity, such as mathematical operations and logic reasoning. If the neural network lacks the ability to model periodicity, it could impair the learning efficiency [Dong et al., 2025b]. From a deeper perspective, periodicity is not just a data feature but reflects a form of structural knowledge — one that allows for the transfer and reuse of abstract rules and principles across different contexts.

# 7 Conclusion and Future Work

In this paper, we have proposed Fourier Analysis Network (FAN), a novel network that addresses periodicity modeling in existing networks while maintaining the general-purpose modeling capability. Experimental results demonstrate that FAN successfully fit both basic and complex periodic functions, whereas other general-purpose networks failed. Moreover, using FAN exhibit clear improvements in real-world tasks, such as symbolic formula representation, time series forecasting, and language modeling, outperforming neural networks such as MLP, KAN, LSTM, Mamba, and Transformer. These promising results, especially the stronger performance and the fewer parameters and FLOPs compared to MLP, suggest its potential to become a key component of foundational models. Some works have demonstrated the superiority of using FAN in diverse tasks, including gravitational wave analysis [Zhao et al., 2024], EEG-based emotion recognition [Wang et al., 2025], and large language modeling [Dong et al., 2025b], etc. In future work, we aim to further broaden the applicability of FAN.

# 8 Acknowledgement

This research is supported by the National Key R&D Program under Grant No. 2023YFB4503801, the National Natural Science Foundation of China under Grant No. 62192733, 62192730, 62192731, the Major Program (JD) of Hubei Province (No.2023BAA024). Moreover, we would like to thank Lecheng Wang and Xuanming Zhang for their participation in discussions related to this work.

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

# A    MLP

The MLP layer $\Phi(x)$ is defined as:

$$\Phi(x) = \sigma(B_m + W_m x), \tag{12}$$

where $B_m \in \mathbb{R}^{d_m}$ and $W_{\bar{p}} \in \mathbb{R}^{d_x \times d_m}$ are learnable parameters with the hyperparameter $d_m$ indicating the first dimension of $W_m$, $\sigma$ denotes the activation function, and MLP can be defined as the stacking of the MLP layer $\Phi(x)$:

$$\text{MLP}(x) = \Phi_L \circ \Phi_{L-1} \circ \cdots \circ \Phi_1 \circ x, \tag{13}$$

where

$$\Phi_l(x) = \begin{cases} \sigma(B_m^l + W_m^l x), & \text{if } l < L, \\ B^L + W^L x, & \text{if } l = L. \end{cases} \tag{14}$$

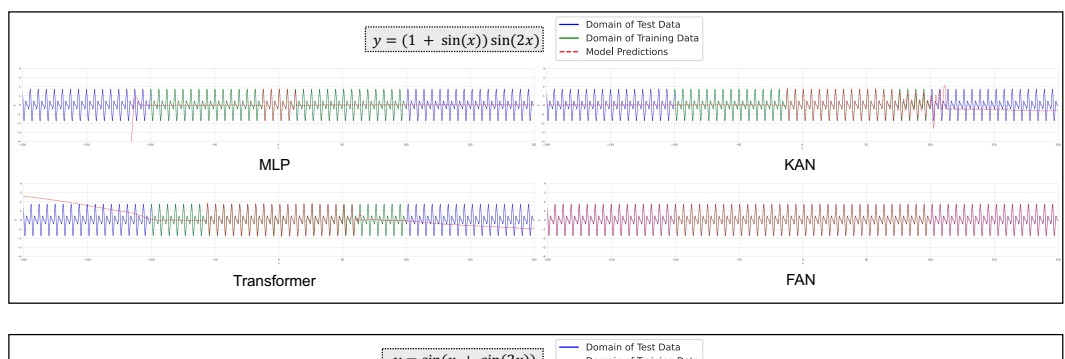

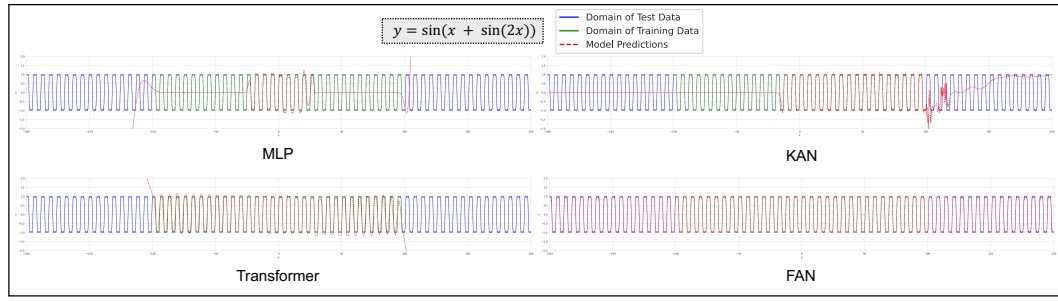

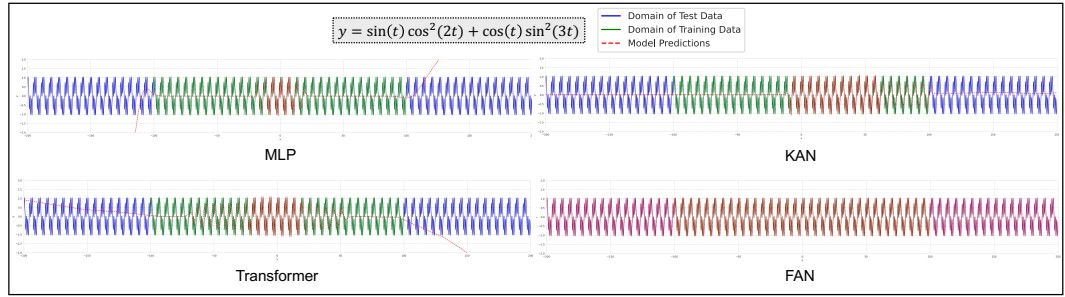

Figure 6: The performance of FAN in periodicity modeling compared to MLP, KAN, and Transformer (Part II), where the green line represents the test data within the domain of training data, while the blue line represents the test data outside the domain of training data.

# B    Additional Experiments

## B.1    Additional Experiments on Periodicity Modeling Tasks.

More experimental results on periodicity modeling tasks are shown in Figure 6.

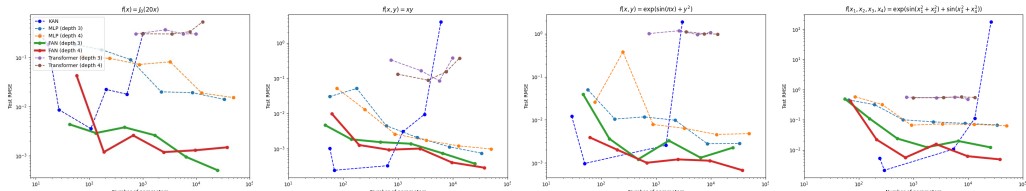

Figure 7: Comparisons of FAN with the baselines, including MLP, KAN, and Transformer, across varying numbers of parameters on symbolic formula representation tasks.

Table 6: Performance of different sequence models on time series forecasting tasks, where Input Length = 96, the **bold** values indicate the lowest value on each row, and Improve means the relative improvements of using FAN and FAN (Gated) based on standard Transformer.

| Dataset | Output Length | LSTM (12.51 M) | | Mamba (12.69 M) | | Transformer (12.12 M) | | Transformer with FAN (11.06 M) | | | |
|---|---|---|---|---|---|---|---|---|---|---|---|
| | | | | | | | | Gated | | Default | |
| | | MSE ↓ | MAE ↓ | MSE ↓ | MAE ↓ | MSE ↓ | MAE ↓ | MSE ↓ | MAE ↓ | MSE ↓ | MAE ↓ |
| Weather | 96 | 1.069 | 0.742 | 0.552 | 0.519 | 0.413 | 0.438 | **0.292** | **0.380** | 0.313 | 0.431 |
| | 192 | 1.090 | 0.778 | 0.700 | 0.595 | 0.582 | 0.540 | 0.535 | 0.550 | **0.472** | **0.525** |
| | 336 | 0.992 | 0.727 | 0.841 | 0.667 | 0.751 | 0.626 | **0.637** | 0.602 | 0.719 | **0.581** |
| | 720 | 1.391 | 0.892 | 1.171 | 0.803 | 0.967 | 0.715 | 0.845 | 0.706 | **0.732** | **0.670** |
| Exchange | 96 | 0.938 | 0.794 | 0.908 | 0.748 | 0.777 | 0.681 | 0.685 | 0.644 | **0.657** | **0.623** |
| | 192 | 1.241 | 0.899 | 1.328 | 0.925 | 1.099 | 0.800 | 0.998 | 0.757 | **0.968** | **0.741** |
| | 336 | 1.645 | 1.048 | 1.512 | 0.992 | 1.614 | 1.029 | 1.511 | 0.961 | **1.266** | **0.905** |
| | 720 | 1.949 | 1.170 | 2.350 | 1.271 | 2.163 | 1.204 | **1.658** | **1.104** | 1.857 | 1.145 |
| Traffic | 96 | 0.659 | 0.359 | 0.666 | 0.377 | 0.656 | 0.357 | 0.647 | 0.355 | **0.643** | **0.347** |
| | 192 | 0.668 | 0.360 | 0.671 | 0.381 | 0.672 | 0.363 | **0.649** | **0.353** | 0.657 | 0.354 |
| | 336 | **0.644** | **0.342** | 0.665 | 0.374 | 0.673 | 0.360 | 0.665 | 0.358 | 0.656 | 0.353 |
| | 720 | **0.654** | **0.351** | 0.662 | 0.364 | 0.701 | 0.380 | 0.682 | 0.369 | 0.673 | 0.363 |
| ETTh | 96 | 0.999 | 0.738 | 0.860 | 0.697 | 1.139 | 0.853 | **0.842** | 0.736 | 0.873 | **0.707** |
| | 192 | 1.059 | 0.759 | **0.849** | **0.700** | 1.373 | 0.932 | 0.885 | 0.748 | 0.914 | 0.741 |
| | 336 | 1.147 | 0.820 | 1.005 | 0.745 | 1.261 | 0.924 | **0.980** | **0.770** | 0.999 | 0.793 |
| | 720 | 1.206 | 0.847 | **0.994** | **0.758** | 1.056 | 0.819 | 1.002 | 0.798 | 1.031 | 0.818 |
| Average (Improve) | – | 1.083 | 0.726 | 1.002 | 0.668 | 0.994 | 0.689 | 0.845 ↓ 15.0% | 0.637 ↓ 7.6% | **0.839** ↓ 15.6% | **0.631** ↓ 8.4% |

## B.2 Additional Experiments on Image Recognition Tasks.

**Image Recognition** is a key computer vision task where image content is identified and categorized. Our evaluation contains four public benchmarks of image recognition: MNIST [LeCun et al., 2010], MNIST-M [Ganin et al., 2016], Fashion-MNIST [Xiao et al., 2017], and Fashion-MNIST-C [Weiss and Tonella, 2022], where MNIST-M and Fashion-MNIST-C are the variants for robustness.

Table 7: Results on image recognition tasks, where OOD Accuracy means the performance on other paired datasets and the **Bold** values indicate the highest values under the same metric.

| Dataset | Accuracy ↑ | | OOD Accuracy ↑ | |
|---|---|---|---|---|
| | CNN | w/ FAN | CNN | w/ FAN |
| MNIST | 99.63 | **99.67** | 28.85 | **30.3** |
| MNIST-M | **94.52** | 94.23 | 82.85 | **83.55** |
| Fashion-MNIST | 94.15 | **94.47** | 49.82 | **51.88** |
| Fashion-MNIST-C | 88.61 | **88.82** | 91.45 | **91.59** |

**Results.** We apply FAN to image recognition tasks on four classic benchmarks, as shown in Table 7. The results show that using FAN outperforms the standard CNN in most cases. We believe that there are also some latent periodic features in image recognition tasks, and FAN's ability to model these periodic features can help CNN achieve competitive or superior performance, especially in OOD scenarios.

## B.3 Evaluation on LLMs with FAN

Table 8 reports the zero-shot results on the LM Eval Harness benchmark. The results show that using FAN outperforms standard Transformer architecture across various tasks with the same training tokens of 200B.

Table 8: Comparison of our approach with well-trained Transformer language models on LM Eval Harness benchmark. Both of them are trained on 200B tokens and using FAN achieves better accuracy.

| Models | arc challenge | arc easy | boolq | hella-swag | open bookqa | piqa | sciq | wino-grande | avg. |
|---|---|---|---|---|---|---|---|---|---|
| Transformer-1B | 29.7 | 63.3 | 59.6 | 52.5 | 34.6 | 71.4 | 85.8 | 55.9 | 56.6 |
| Ours | 32.1 | 63.5 | 60.1 | 53.8 | 34.7 | 72.5 | 89.9 | 56.1 | 57.9 |

## B.4 FAN for Solving SciML Problems

We conduct experiments on the SciML problem that includes the Fourier function class following the work [Li et al., 2021]. The Burgers' equation, a non-linear partial differential equation, is frequently used in scientific computing to model shock waves and traffic flow, among other phenomena. The detailed error rate on Burgers' equation is listed in the Table 9. We can find that replacing the MLP Layer with FAN Layer in Fourier Neural Operator (FNO) [Li et al., 2021] can achieve clear improvements on each setting of resolution $s$ of this task.

Table 9: The error rate on Burgers' equation. The values in the table represent the Average Relative Error for Burgers' equation with lower values indicating better performance.

| Model | $s = 256$ | $s = 512$ | $s = 1024$ | $s = 2048$ | $s = 4096$ | $s = 8192$ |
|---|---|---|---|---|---|---|
| FNO | 5.93% | 6.14% | 6.03% | 6.75% | 7.36% | 9.93% |
| FNO with FAN | **5.26%** | **5.17%** | **5.18%** | **6.73%** | **6.35%** | **7.06%** |

## B.5 Comparison with Frequency-based Models in Time Series Forecasting Tasks

To compare with frequency-based models in Time Series Forecasting tasks such as FEDformer [Zhou et al., 2022], we replace MLP with FAN in frequency-based models. We present the experimental results in Table 10, where the results of FEDformer are cited from its paper directly. From the results, we can find that FEDformer with FAN can outperform FEDformer in almost all cases.

Table 10: Results of comparison with frequency-based models in time series forecasting tasks.

| Dataset | Len | FEDformer | | with FAN | |
|---|---|---|---|---|---|
| | | MSE | MAE | MSE | MAE |
| Traffic | 96 | 0.587 | 0.366 | **0.577** | **0.357** |
| | 192 | 0.604 | 0.373 | **0.601** | **0.366** |
| | 336 | 0.621 | 0.383 | **0.620** | **0.378** |
| | 720 | 0.626 | 0.382 | **0.619** | **0.370** |
| Exchange | 96 | 0.148 | 0.278 | **0.138** | **0.267** |
| | 192 | 0.271 | 0.380 | **0.261** | **0.371** |
| | 336 | **0.460** | **0.500** | 0.461 | 0.503 |
| | 720 | 1.195 | 0.841 | **1.159** | **0.827** |
| Electricity | 96 | 0.193 | 0.308 | **0.184** | **0.298** |
| | 192 | 0.201 | 0.315 | **0.199** | **0.313** |
| | 336 | 0.214 | 0.329 | **0.212** | **0.325** |
| | 720 | 0.246 | 0.355 | **0.239** | **0.347** |

### B.6 Comparison with Directly Learning the Coefficients

We compare FAN with a baseline of directly learning the coefficients, which inputs $sin(x)$ and $cos(x)$ and then uses the MLP Layer instead of the FAN Layer to model the Fourier coefficients. In this setting, frequencies are fixed and only the coefficients are learned, which may limit the model's ability to capture patterns not aligned with these frequencies. Taking simple $f(x) = x \bmod 5$ as an example, this setting may not even converge at all, because the frequency of $x \bmod 5$ is inconsistent with $sin(x)$ and $cos(x)$. The experimental results of their loss are shown in Table 11.

Table 11: Comparison of FAN and directly learning the coefficients on fitting $f(x) = x \bmod 5$.

| Epoch | 50 | 100 | 150 | 200 |
|---|---|---|---|---|
| Directly learning the coefficients | 2.10 | 2.09 | 2.09 | 2.08 |
| FAN | 0.28 | 0.23 | 0.18 | 0.17 |

### B.7 The influence of hyperparameters $d_p$

We evaluate the influence of hyperparameters $d_p$ as shown in Figure 8.

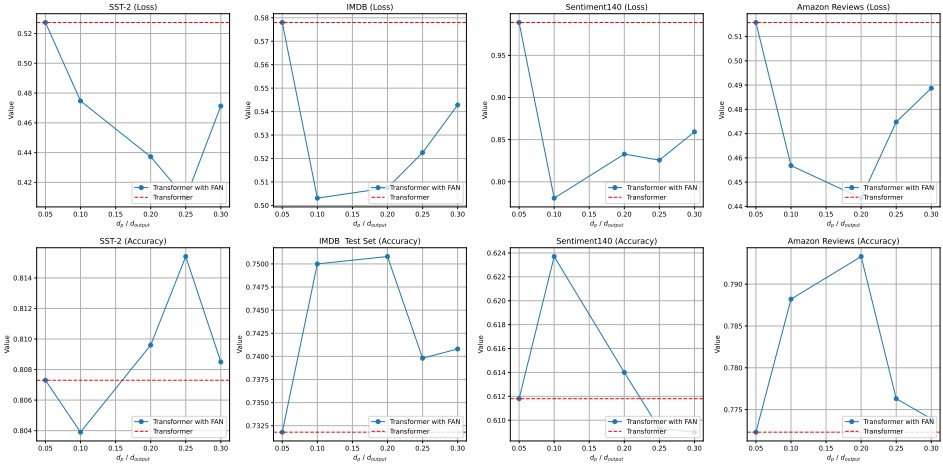

Figure 8: The influence of hyper-parameters $d_p$ on language modeling tasks. We use the red dashed line to represent the performance of the standard Transformer.

### B.8 The effectiveness of the FAN Layer for deep neural networks

We evaluate the effect of varying the number of FAN layers from 3 to 20 on periodicity modeling tasks, employing residual connections to mitigate overfitting. The experimental results show that both the best training loss and test loss still decrease slowly as the number of layers increases.

Furthermore, on Language Modeling tasks, we replaced 24 MLP Layers of Transformer with 24 FAN Layers, i.e. Transformer with FAN, and it also achieved clear improvements on each task, especially for OOD zero-shot evaluation scenarios. These findings indicate that FAN Layer is effective for deep neural networks.

### B.9 Experiments on Time Series Forecasting with Instance Normalization

We conduct experiments on time series forecasting tasks with instance normalization [Ulyanov et al., 2016], and the results are shown in Table 12. We find that applying instance normalization before the architecture can effectively improve the performance.

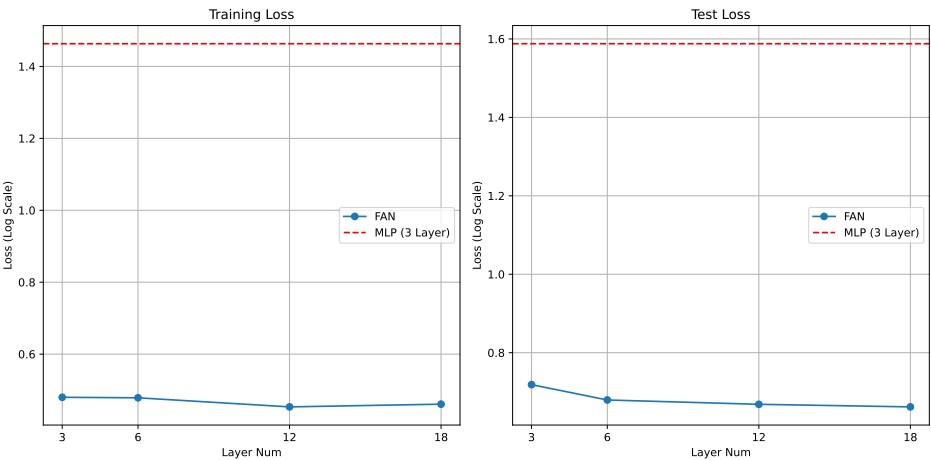

Figure 9: Performance of Deeper FAN on fitting $y = e^{\sin^2(\pi x)+\cos(x)+(x \mod 3)} - 1$.

Table 12: Results on time series forecasting tasks with instance normalization, where Input Length = 96, the **bold** values indicate the lowest value on each row, and the improve means the relative improvements of using FAN and FAN (Gated) based on Transformer.

| Dataset | Output Length | Transformer (12.12 M) | | Transformer with FAN (11.06 M) | | | |
| | | | | Gated | | Default | |
| | | MSE ↓ | MAE ↓ | MSE ↓ | MAE ↓ | MSE ↓ | MAE ↓ |
|---|---|---|---|---|---|---|---|
| Weather | 96 | 0.1772 | 0.2301 | 0.1864 | 0.2352 | **0.1756** | **0.2247** |
| | 192 | 0.2438 | 0.2844 | 0.2445 | 0.2834 | **0.2327** | **0.2760** |
| | 336 | **0.3077** | **0.3267** | 0.3156 | 0.3320 | 0.3118 | 0.3291 |
| | 720 | 0.4253 | 0.3982 | **0.3909** | **0.3782** | 0.4113 | 0.3906 |
| Exchange | 96 | 0.1433 | 0.2653 | **0.1157** | **0.2452** | 0.1436 | 0.2666 |
| | 192 | 0.2563 | **0.3552** | **0.2539** | 0.3611 | 0.2651 | 0.3757 |
| | 336 | 0.5273 | 0.5218 | **0.4329** | **0.4891** | 0.5092 | 0.5326 |
| | 720 | 1.7401 | 0.9273 | 1.5783 | 0.9303 | **1.0599** | **0.7657** |
| Traffic | 96 | 0.6160 | 0.3449 | **0.6030** | 0.3334 | 0.6109 | **0.3319** |
| | 192 | 0.6329 | 0.3479 | 0.6239 | 0.3404 | 0.6258 | **0.3370** |
| | 336 | 0.6369 | 0.3485 | 0.6416 | 0.3487 | **0.6200** | **0.3380** |
| | 720 | 0.6555 | 0.3577 | 0.6645 | 0.3574 | **0.6412** | **0.3525** |
| ETTh | 96 | 0.3881 | **0.4097** | 0.4082 | 0.4292 | **0.3833** | 0.4149 |
| | 192 | 0.5766 | 0.4999 | **0.4695** | **0.4514** | 0.5039 | 0.4640 |
| | 336 | 0.5782 | 0.5100 | 0.5556 | 0.5012 | **0.5417** | **0.4940** |
| | 720 | 0.5841 | 0.5230 | **0.5070** | **0.4943** | 0.5272 | 0.4951 |
| Average (Improve) | – | 0.531 | 0.416 | 0.499 ↓ 6.1% | 0.406 ↓ 2.2% | **0.472** ↓ 11.0% | **0.399** ↓ 4.1% |

## B.10 Layer-wise Spectral Analysis

We conduct experiments on layer-wise spectral analysis below. We perform a Fast Fourier Transform (FFT) on each layer's outputs and calculate four key metrics to quantify the spectral characteristics:

1. **Spectral Centroid:** Measures the "center of mass" of the spectrum, indicating whether the layer's features are concentrated in low or high-frequency regions.

2. **Spectral Sparsity (L1/L2 Norm):** Quantifies how concentrated the spectral energy is within a few frequency bins. A higher value implies a more structured and less noisy signal.

3. **Spectral Entropy:** Measures the uniformity and predictability of the spectrum. A lower entropy indicates a more ordered and well-defined spectral structure.

4. **Dominant Energy Ratio (Top-5):** The proportion of total spectral energy contained within the top 5 most dominant frequency components, indicating how focused the representation is on key periodic features.

The results reveal a highly effective multi-stage learning process, which is more sophisticated than a simple monotonic evolution of frequencies. We observe a clear three-stage "Deconstruction-Exploration-Reconstruction" mechanism:

1. **Initial Approximation (Layer 1):** The first layer rapidly forms an initial, highly-focused approximation of the signal, as shown by its very high Dominant Energy Ratio (96.1%).

2. **Feature Deconstruction and Exploration (Layers 2–8):** To model the function's complex, non-sinusoidal components (especially the $x \pmod 3$ term, which requires a wide range of Fourier series terms), the intermediate layers must first "deconstruct" the signal. This is evidenced by a sharp increase in Spectral Entropy and a decrease in the Dominant Energy Ratio. The network actively disperses energy across a broader spectrum to explore and capture these challenging features, showcasing the flexibility afforded by its depth.

3. **Integration and Reconstruction (Layers 9–11):** In the final layers, the model's task shifts from exploration to integration. It "reconstructs" a final, efficient representation from the features learned in the middle layers. This is marked by a dramatic decrease in both Spectral Entropy and Spectral Centroid, alongside a sharp increase in the Dominant Energy Ratio to a final value of 93.8%. The network converges to a "clean", low-frequency, and highly structured representation that is optimal for the final linear layer to map to the target output.

Table 13: Layer-wise spectral analysis of FAN layer outputs.

| Layer | Spectral Centroid | Spectral Sparsity | Spectral Entropy | Dominant Energy Ratio (Top-5) |
|---|---|---|---|---|
| FAN Layer 1 | 4.1213 | 3.4767 | 1.2264 | 0.9612 |
| FAN Layer 2 | 2.8760 | 5.0003 | 3.2549 | 0.7602 |
| FAN Layer 3 | 2.8804 | 5.0626 | 3.1556 | 0.7807 |
| FAN Layer 4 | 2.8810 | 4.7149 | 2.6616 | 0.8426 |
| FAN Layer 5 | 3.0820 | 4.5832 | 2.2248 | 0.8753 |
| FAN Layer 6 | 3.0815 | 5.2388 | 2.5560 | 0.8378 |
| FAN Layer 7 | 2.6955 | 5.8367 | 3.0115 | 0.7806 |
| FAN Layer 8 | 2.9132 | 5.5387 | 2.7301 | 0.8086 |
| FAN Layer 9 | 2.7376 | 4.1371 | 1.6760 | 0.8986 |
| FAN Layer 10 | 2.1266 | 3.1509 | 1.0673 | 0.9356 |
| FAN Layer 11 | 1.7721 | 2.9775 | 0.9270 | 0.9375 |

## B.11 Ablation Study

We conduct ablation studies on just cosine function, having FAN layers only in part of the network, and freezing $W_p$. The results show that FAN demonstrates a clear advantage over the variants in Periodicity Modeling and Language Modeling tasks.

Table 14: Results for ablation studies on the Periodicity Modeling task.

| Periodicity Modeling | Epoch=0 | | Epoch=100 | | Epoch=1000 | |
|---|---|---|---|---|---|---|
| | training loss | test loss | training loss | test loss | training loss | test loss |
| FAN_cos | 39.85 | 63.42 | **2.67** | 10.18 | 1.80 | 5.26 |
| FAN_replace_first_1/3_part | 39.54 | **46.15** | 2.95 | **6.81** | 1.37 | 45.44 |
| FAN_replace_last_1/3_part | 42.82 | 55.46 | 21.96 | 27.86 | 22.79 | 30.51 |
| freezing $W_p$ for FAN | 40.52 | 60.20 | 15.57 | 89.09 | 1.13 | 156.25 |
| FAN | **39.62** | 61.02 | 2.75 | 7.43 | **1.05** | **4.15** |

Table 15: Results for ablation studies on the Language Modeling task.

| Language Modeling | Train Loss | In-domain Test Loss | OOD Test Loss |
|---|---|---|---|
| FAN_cos | 0.2419 | 0.4802 | 0.7727 |
| FAN_replace_first_1/3_part | 0.2693 | 0.4313 | 0.6700 |
| FAN_replace_last_1/3_part | **0.2417** | 0.4660 | 0.8052 |
| freezing $W_p$ for FAN | 0.2376 | 0.4736 | 0.6324 |
| FAN | 0.2434 | **0.4094** | **0.6077** |

## C    Experimental Details

**Baselines.**    In our experiments, we mainly compare FAN with the following baselines: 1) **MLP** [Rosenblatt, 1958], 2) **Transformer** [Vaswani et al., 2017], 3) **KAN** [Liu et al., 2024], 4) **LSTM** [Hochreiter and Schmidhuber, 1997], 5) **Mamba** [Gu and Dao, 2023], 6) **CNN** [LeCun et al., 1998]. Details of the baselines are given in Appendix F. Moreover, we also include the following variants of FAN into our comparisons: I) **FAN (Gated)**: a variant of FAN that adds gates to control the tendency of the layer, with the formula defined as $\phi_g(x) = [g \cdot \cos(W_p x) || g \cdot \sin(W_p x) || (1-g) \cdot \sigma(B_{\bar{p}} + W_{\bar{p}} x)]$, where $g$ is a learnable parameter. II) **Transformer with FAN and Transformer with FAN (Gated)**: we replace each MLP layer in Transformer with the FAN layer computed via Eq. (9) and the layer of FAN (Gated), respectively. III) **CNN with FAN**: similarly, we replace each MLP layer in CNN with the FAN layer.

### C.1    Implementation Details.

We conduct our experiments on a single GPU of Tesla A100-PCIe-40G. Unless otherwise specified, we use the following hyperparameters in the experiments. The model architecture consists of 3 to 24 layers, the activation function $\sigma$ is set to GELU [Hendrycks and Gimpel, 2016], and the dimension of the projection matrix $W_p$ is set to $d_p = \frac{1}{4} d_h$, where $d_h$ denotes the dimension of the hidden layers. We employ the AdamW optimizer [Loshchilov and Hutter, 2019] for the model's training process.

### C.2    Setup of Periodicity Modeling

In periodicity modeling tasks, FAN, MLP, and KAN each consist of three layers with comparable FLOPs, while the Transformer model comprises twelve layers. For consistency, we set the hidden layer dimension ($d_h$) to 2048 for FAN, MLP, and Transformer. In the case of KAN, we follow its original paper [Liu et al., 2024], where the spline order ($K$) and the number of spline intervals ($G$) are set to 3 and 50, respectively. We apply a learning rate of $1 \times 10^{-5}$ for training all models. We ensured that the data density of each period in tasks was consistent, meaning that each cycle contained a fixed quantity of 10,000 training data points.

### C.3    Setup of Symbolic Formula Representation

In symbolic formula representation tasks, we used the create_dataset function from the official KAN repository to generate the datasets. Each dataset contains 3000 training samples and 1000 test samples, with all input variables randomly sampled from the range [-1, 1]. We followed the training settings from the original KAN paper, training all methods using LBFGS and Adam for 1800 steps, and selecting the best-performing result from the two optimization approaches. For KAN, we increased the number of grid points to scale up the parameter size, covering $G = \{3, 5, 10, 20, 50, 100, 200, 500, 1000\}$. For other methods, we scaled up the parameter size by increasing the number of layers and the dimensions of hidden layers.

### C.4    Setup of Time Series Forecasting

In time series forecasting task, we implement our model based on the codebase by [Wu et al., 2021]. Each model comprises 2 encoder layers and 1 decoder layer. We fix the hidden size for both the Transformer and our model at 512, with the feedforward dimension set to 2048 (four times the hidden size). The parameter sizes detailed in the main text correspond to the Exchange dataset; variations in

the number of variables across different datasets influence the linear layers in the model. We adjust the hidden sizes of the other models to align with the Transformer parameters for fairness.

## C.5 Setup of Language Modeling

In language modeling task, we employ the BERT tokenizer [Devlin et al., 2018] and an embedding layer with a dimensionality of 768, except for Mamba, which adheres to its default settings as specified in the original paper [Gu and Dao, 2023]. The architecture features 4, 24, and 12 layers with hidden sizes of 1800, 768, and 768 for LSTM, Mamba, and Transformers, respectively. To mitigate training stagnation in deeper LSTM models, we reduce the number of layers while increasing the hidden size to balance the parameters. Importantly, Mamba's layer count is twice that of a similarly sized Transformer, as each layer consists of two Mamba blocks (Multihead attention block + MLP block).

## C.6 Setup of Image Recognition

In image recognition tasks, we employ the CNN as the baseline model, which consists of four Convolutional Layers and two MLP Layers (It achieves a 0.37% error rate on MNIST without augmentation, outperforming the SOTA CNN's 0.63% [Wan et al., 2013]). We replace MLP with FAN in CNN, i.e., CNN with FAN, as the counterpart, ensuring that they have similar parameters. For each task, we use stochastic gradient descent with momentum (SGDM) as the optimizer, the learning rate is set to 0.01, and the training process runs for 100 epochs.

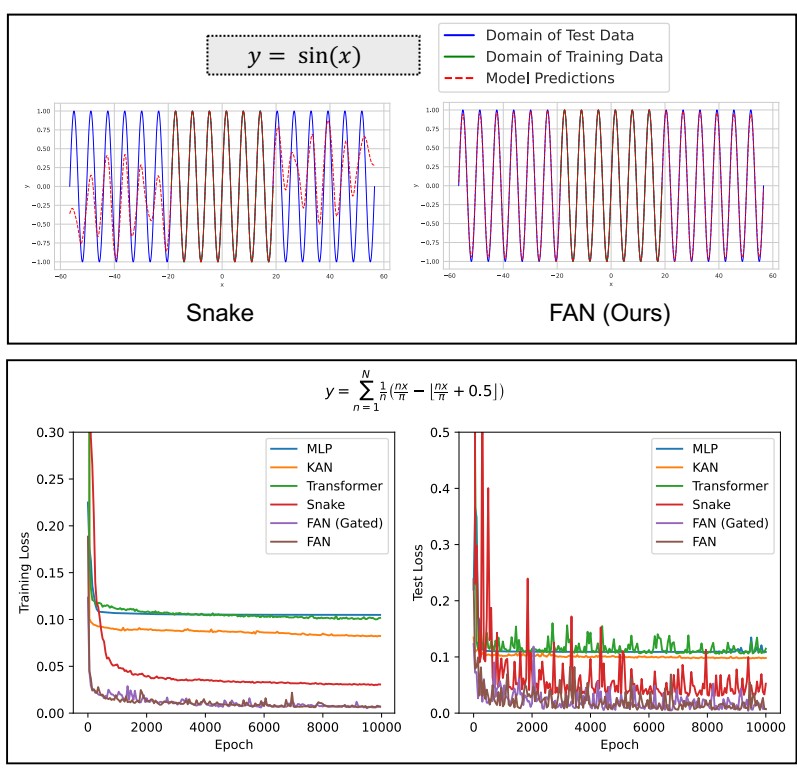

Figure 10: Comparisons of FAN with MLP (Snake) [Liu et al., 2020] in fitting periodic functions.

# D    Comparison of FAN and Snake Activation Function

We compare FAN with Snake, a previous approach used for improving the fitting of periodic functions with neural networks. The results are shown in Figure 10.

# E    Compliance with the Universal Approximation Theorem

The Universal Approximation Theorem asserts that a feed-forward network with a single hidden layer, containing a sufficiently large and finite number of neurons, can approximate any continuous function defined on compact subsets of $\mathbb{R}^n$, provided that the activation function is non-constant, continuous, and nonlinear. In the case of the Fourier Analysis Network (FAN) layer, we define the mapping as:

$$\phi(x) = \left[ \cos(W_p x) \, \middle\| \, \sin(W_p x) \, \middle\| \, \sigma(B_{\bar{p}} + W_{\bar{p}} x) \right],$$

where $\|$ denotes concatenation, and $\sigma(\cdot)$ represents a standard nonlinear activation function, such as ReLU or GELU. The components $\cos(W\_px)$ and $\sin(W\_px)$ are non-constant, continuous, and nonlinear functions, satisfying the requisite conditions for an activation function in the Universal Approximation Theorem. Therefore, the FAN layer conforms to the Universal Approximation Theorem, enabling it to approximate arbitrary continuous functions on compact subsets of $\mathbb{R}^n$.

This proof demonstrates that the FAN layer, through its periodic components (sine and cosine functions) and the nonlinear activation $\sigma(\cdot)$, satisfies the key conditions of the Universal Approximation Theorem, ensuring its capability to approximate complex functional mappings.

# F    More Details of Baselines

In our experiments, we mainly compare FAN with the following baselines. 1) **MLP** [Rosenblatt, 1958]: the most classic model, which is widely used in the backbone of various models. 2) **Transformer** [Vaswani et al., 2017]: a prevalent model known for its self-attention mechanism, which achieves outstanding performance on various tasks. 3) **KAN** [Liu et al., 2024]: an emerged model specialized for symbolic formula representation, which uses the b-spline functions instead of fixed activation functions. 4) **LSTM** [Hochreiter and Schmidhuber, 1997]: a well-known recurrent neural network (RNN) that can capture long-term dependencies on sequential data. 5) **Mamba** [Gu and Dao, 2023]: an emerged selective state space model (SSM) that achieves competitive performance on some tasks with sequential inputs. 6) **CNN** [LeCun et al., 1998]: convolutional neural network contains the convolutional layers, which are effective in processing image data.

For Fourier-based Networks, we mainly compare FAN with 1) Fourier Neural Network (FNN) [Silvescu, 1999] using the cosine or sine function or their linear combinations as the activation function, such as SIREN [Sitzmann et al., 2020]. 2) Fourier Series Neural Network (FSNN) is defined as Eq. (3), which shares the parameters and computation of Sin and Cos part. 3) Fourier Transform Neural Network (FTNN) is a type of neural network that employs Fourier Transform to process the intermediate output in the neural network, such as FNO [Li et al., 2021].

# G    More Detailed Discussion with Fourier-based Neural Network

For FNNs [Silvescu, 1999, Liu, 2013, Parascandolo et al., 2016, Uteuliyeva et al., 2020], they face challenges in scaling to deeper networks, i.e., the capacity of their deep networks to fit the Fourier coefficients is independent of the network depth, as analyzed in Section 3. The depth scalability limits their applicability to more complex, general-purpose tasks such as language modeling. Our core differences are, "we design FAN based on the following principles: 1) the capacity of FAN to represent the Fourier coefficients should be positively correlated to its depth; 2) the output of any hidden layer can be employed to model periodicity using Fourier Series through the subsequent layers." In Section 4.3, we conduct experiments to compare our approach with FNNs, and FNNs cannot fit language modeling tasks, but our approach works well. We provide the analysis of FNNs compared to FAN below. We mainly discuss the work [Silvescu, 1999, Liu, 2013, Parascandolo et al., 2016], due to the work [Uteuliyeva et al., 2020] is a comparative study without proposing a new method.

For work [Lee et al., 2021, Belcak and Wattenhofer, 2022], they focus on different purposes from our work. And work [Lee et al., 2021] assumes all input signals have the period of 1 (as stated in page 3 of its paper), which we conducted experiments on the same setting in Appendix B.6, and it cannot fit our periodicity modeling tasks.

Table 16: Comparison of parameters and FLOPs for different layers, where $d_i = d_{\text{input}}$ (Input dimension hyperparameter), $d_o = d_{\text{output}}$ (Output dimension hyperparameter), $d_p$ = FAN layer hyperparameter (default $\frac{1}{4}d_o$), $d_h$ = Hidden dimension hyperparameter, $d_c, d_s$ = Layer hyperparameters of cosine/sine branch dimensions, $m$ = Layer hyperparameter of projections number, $\gamma$ = FLOPs per nonlinear activation ($\sigma$, cos, or sin).

| Metric | FAN Layer | Layer of [Silvescu, 1999] | Layer of [Liu, 2013] | Layer of [Parascandolo et al., 2016] |
|---|---|---|---|---|
| Formula | $[\cos(W_p x) \parallel \sin(W_p x) \parallel \sigma(B_{\bar{p}} + W_{\bar{p}} x)]$ | $W_f \prod_m \cos(W_{a_m} x + B_{a_m}) + B_f$ | $W_{f_c} \cos(W_{a_c} x + B_{a_c}) + B_{f_c} + W_{f_s} \sin(W_{a_s} x + B_{a_s}) + B_{f_s}$ | $W_f \sin(W_a x + B_a) + B_f$ |
| Num Params | $(1 - \frac{d_p}{d_o})(d_i d_o + d_o)$ | $m(d_i d_h + d_h) + d_o d_h + d_o$ | $d_i(d_c + d_s) + (d_c + d_s) + d_o(d_c + d_s) + 2d_o$ | $d_i d_h + d_h + d_o d_h + d_o$ |
| FLOPs | $(1 - \frac{d_p}{d_o}) \times 2d_i d_o + \gamma d_o$ | $2m d_h d_i + d_h(m-1) + 2d_h d_o + \gamma m d_h$ | $2d_i(d_c + d_s) + 2d_o(d_c + d_s) + \gamma(d_c + d_s)$ | $2d_h(d_i + d_o) + \gamma d_h$ |

## G.1 Limitation

First, we only demonstrate the effectiveness of FAN on some mainstream real-world tasks (including symbolic formula representation, time series forecasting, language modeling, image recognition, etc.), and we aim to further broaden the applicability of FAN in our future work. Second, although we have explored the generalizability of FAN and confirmed that FAN outperforms the baseline method in some real-world tasks, the boundaries of this model's generalizability remain unknown. However, we have not yet identified specific scenarios where it performs poorly. We leave this for our future work.

