# OpenReview forum: "FAN: Fourier Analysis Networks"
_NeurIPS.cc/2025/Conference — NeurIPS 2025 poster_

### Official Review · Reviewer_rvp8 · 2025-06-30

**Clarity:** 4
**Significance:** 2
**Originality:** 3
**Rating:** 4
**Confidence:** 2

**Summary:**

The paper addresses the limitation of traditional neural networks that can’t properly model periodic functions. The paper introduces Fourier Analysis Networks (FAN), a novel network architecture, based on Fourier analysis, that can replace MLP blocks so models can better represent periodic structure with fewer parameters and FLOPs. The paper shows the effectiveness of FAN, compared to other baseline architectures like MLP, KAN and Transformer on a variety of periodic modelling tasks, time series forecasting and sentiment analysis tasks.

**Questions:**

I believe FAN represents a valuable contribution for modeling periodic functions, but I struggle to see how it qualifies as a general-purpose architecture given the experimental evidence. Could the authors clarify their vision for FAN's practical applications? Specifically, which real-world tasks would benefit from the Fourier-based inductive bias, and how would you recommend identifying when FAN layers would be preferable to standard MLPs? While I understand the authors' argument that many machine learning tasks may benefit from FAN due to hidden periodicity, there's still a gap between this theoretical vision and the experimental validation.


Minor point:
To improve clarity, I’d suggest moving the description of the FAN(Gated) variant to the main paper.

**Ethical Concerns:**

["NO or VERY MINOR ethics concerns only"]

**Final Justification:**

The authors addressed my concerns about the lack of experimental evidence to support the claim of FAN being a general-purpose model. They provided a better explanation for their motivation and pointed to additional experiments in the Appendix. Although I still believe that this claim of general-purpose capabilities is a bit bold and requires further experimentation, I agree that it can be left for future work, especially since the authors promised to rephrase their statements and add discussion of limitations. I'm increasing my score accordingly.

**Limitations:**

I'd recommend the authors include a more thorough discussion of the limitations in their real-world task selection, specifically addressing why certain tasks were chosen and which types of applications might not benefit from FAN.

**Quality:**

3

**Strengths And Weaknesses:**

## Strengths
- The writing is clear and the motivation is well explained with experiments showing that standard networks extrapolate poorly in periodic data.
- The proposed architecture embeds efficiently periodicity while reducing the parameter count.
- The experimental results on periodic tasks are convincing and the method clearly improves OOD generalization.

## Weaknesses
My main concern is that the paper claims FAN to be a general-purpose model, while lacking extensive experiment analysis to support this claim. The paper compares against vanilla architectures rather than specialized state-of-the-art models. If FAN layers can serve as replacements for MLPs in existing architectures, would current specialized SOTA actually benefit from this substitution? Sentiment analysis seems like an odd choice to showcase FAN's language modeling benefits, there is no clear periodic structure that Fourier analysis would capture. Combined with the lack of comparison against SOTA models this experiment fails to demonstrate any meaningful advantage for real-world language tasks.

---

> ### Author Rebuttal · Authors · 2025-07-31
>
> Thank you for your insightful comments. We will respond to your comment point-by-point below.
>
> > I believe FAN represents a valuable contribution for modeling periodic functions, but I struggle to see how it qualifies as a general-purpose architecture given the experimental evidence.
>
> Thanks for recognizing the value of our work. Compared with existing Fourier-based networks, which are typically designed for specific tasks, we have explored the capability of FAN to be applied to more types of tasks. Based on the current experimental results, we observed that FAN has the potential to be applied to a range of real-world tasks. However, whether FAN equips the general-purpose capabilities remains a question to be answered. We will adjust the statements like "maintains general-purpose modeling capability" to more precise descriptions, such as "maintains capability to be applied to more types of tasks" in our revised manuscript.
>
> > Could the authors clarify their vision for FAN's practical applications? Specifically, which real-world tasks would benefit from the Fourier-based inductive bias, and how would you recommend identifying when FAN layers would be preferable to standard MLPs? &  I'd recommend the authors include a more thorough discussion of the limitations in their real-world task selection, specifically addressing why certain tasks were chosen and which types of applications might not benefit from FAN.
>
> That's a good question. Periodicity is a fundamental and ubiquitous characteristic in both the natural sciences and human activities. However, existing general-purpose neural networks, such as MLPs and Transformers, exhibit inherent limitations in effectively capturing and modeling these periodic features, while existing Fourier-based networks face challenges in scaling to deeper networks and are typically designed for specific tasks. To this end, we propose FAN to effectively address periodicity modeling challenges while maintaining capability to be applied to more types of tasks.
>
> Various forms of real-world data contain either explicit or implicit periodic patterns. We believe that these real-world tasks will benefit from FAN. When the task is completely aperiodic, FAN degenerates to MLP. Although we have explored the generalizability of FAN and confirmed that FAN outperforms the baseline method in some mainstream real-world tasks (including symbolic formula representation, time series forecasting, language modeling, image recognition, etc.), the boundaries of this model's generalizability remain unknown. However, we have not yet identified specific scenarios where it performs poorly. We leave this for our future work. We have added this discussion to our revised manuscript.
>
> >  The paper compares against vanilla architectures rather than specialized state-of-the-art models. If FAN layers can serve as replacements for MLPs in existing architectures, would current specialized SOTA actually benefit from this substitution?
>
> Yes, we hope it can replace MLP in some aspects, and we attempted to demonstrate this in our paper (due to space and experimental limitations, we only explored its advantages on existing tasks). As shown in Appendix B.3 and B.4, we compared SOTA methods with using FAN in Large Language Modeling scenario and Solving SciML Problems scenario, where it outperforms specialized SOTA methods.
>
> > Sentiment analysis seems like an odd choice to showcase FAN's language modeling benefits.
>
> Thanks for your comment. Our intention with the experiments was twofold. First, our primary goal was to validate FAN's effectiveness in the fundamental task of language modeling, which we demonstrated through its ability to achieve a lower entropy loss. This part of the experiment directly addresses its language modeling capabilities. Second, we wanted to conduct an exploratory investigation into whether FAN has the potential to be applied to higher-level, downstream NLP tasks, such as sentiment analysis (one of the most widespread real-world NLP applications, used for analyzing user reviews, social media sentiment, and brand monitoring). If you think this result is odd, we are happy to remove the sentiment analysis results, with only maintaining the results of language modeling, in the camera-ready version.
>
> > To improve clarity, I’d suggest moving the description of the FAN(Gated) variant to the main paper.
>
> Thank you for your suggestion! We have moved the description of the FAN(Gated) variant to the main paper in our revised version.

---

> ### Comment · Reviewer_rvp8 · 2025-08-02
>
> Thank you for addressing my concerns! I have a better understanding now of why you see FAN as a general replacement for MLP. About the sentiment analysis experiment, it is an interesting result to include in the main paper, but it's also interesting to see how the method performs on other LLM tasks. I'd suggest directly referring readers to Appendix B.3 for more results. Overall, I agree that vast analysis of FAN performance on general tasks can be conducted in future work. I increased my score.

---

> ### Author Response · Authors · 2025-08-04
>
> Dear Reviewer rvp8,
>
> Thank you for your encouraging feedback and constructive suggestions! We sincerely appreciate the time and effort you’ve dedicated to reviewing our work. Your insightful comments have greatly strengthened the paper, and we’re grateful for your support in raising the score.
>
> Best regards,
> Authors

---

### Official Review · Reviewer_GT2m · 2025-07-03

**Clarity:** 3
**Significance:** 3
**Originality:** 2
**Rating:** 5
**Confidence:** 3

**Summary:**

The authors propose Fourier Analysis Networks (FAN), a replacement for MLP blocks that concatenates $cos(W x)$, $sin (W x)$ and a conventional non-linear projection inside every hidden layer. This design is driven by two principles: (1) depth-scaled capacity to learn Fourier coefficients and (2) periodicity accessibility at any hidden layer. The idea is that this should enable FAN to capture periodic patterns effectively while still remaining applicable to a broad range of tasks. They obtain consistent gains on experiments from a variety of domains, including symbolic formula representation, time-series forecasting and language modeling.

**Questions:**

Would it be possible to do a layer-wise spectral analysis to illustrate how coefficients/frequencies evolve with depth?

Do FAN’s cosine/sine blocks hamper tasks that are intrinsically non-periodic? A negative result would be informative, and in my opinion, it would not reduce the value of the contribution.

Section on prior work says: "Previous studies have explored Fourier-based networks, but these networks generally perform well on specific tasks, while their performance on more general tasks tends to be poorer" and lists a number of citations. It would be informative to expand that sentence and section further (perhaps in the appendix) and detail more precisely how the proposed approach differs from the prior work.

I suggest doing some ablation experiments to better understand what are the key contributors in FAN. For instance, freezing W_p or having just sine or cosine function or having FAN layers only in part of the network.

**Ethical Concerns:**

["NO or VERY MINOR ethics concerns only"]

**Final Justification:**

The authors have addressed my comments reasonably well. There are a couple more of small suggestions remaining. Overall, I think the work is interesting and relevant.

**Limitations:**

Yes.

**Paper Formatting Concerns:**

All good.

**Quality:**

3

**Strengths And Weaknesses:**

The approach seems to be novel in the space of Fourier Series NNs, specifically in how it integrates sine and cosine functions with other activation functions.

The variety and scope of experiments are extensive and show impressive generalization across domains.

The code is provided and in general, the idea is relatively straightforward so it should be reproducible.

Additional analyses of layer activations and ablations could help to better understand why the approach works better than existing Fourier Series NNs.

---

> ### Author Rebuttal · Authors · 2025-07-31
>
> Thank you for your thoughtful comment and constructive feedback. We will respond to your comment point-by-point below.
>
> > Additional analyses of layer activations and ablations could help to better understand why the approach works better than existing Fourier Series NNs. Would it be possible to do a layer-wise spectral analysis to illustrate how coefficients/frequencies evolve with depth?
>
> Thanks for your suggestions. We conduct experiments on layer-wise spectral analysis and have added the experimental results to our revised manuscript.  We perform a Fast Fourier Transform (FFT) on each layer's outputs and calculate four key metrics to quantify the spectral characteristics:
> 1)  Spectral Centroid: Measures the "center of mass" of the spectrum, indicating whether the layer's features are concentrated in low or high-frequency regions.
> 2)  Spectral Sparsity (L1/L2 Norm): Quantifies how concentrated the spectral energy is within a few frequency bins. A higher value implies a more structured and less noisy signal.
> 3)  Spectral Entropy: Measures the uniformity and predictability of the spectrum. A lower entropy indicates a more ordered and well-defined spectral structure.
> 4)  Dominant Energy Ratio (Top-5): The proportion of total spectral energy contained within the top 5 most dominant frequency components, indicating how focused the representation is on key periodic features.
>
> The results reveal a highly effective multi-stage learning process, which is more sophisticated than a simple monotonic evolution of frequencies. We observe a clear three-stage "Deconstruction-Exploration-Reconstruction" mechanism:
>
> 1. Initial Approximation (Layer 1): The first layer rapidly forms an initial, highly-focused approximation of the signal, as shown by its very high Dominant Energy Ratio (96.1%).
>
> 2. Feature Deconstruction and Exploration (Layers 2-8): To model the function's complex, non-sinusoidal components (especially the x mod 3 term, which requires a wide range of Fourier series terms), the intermediate layers must first "deconstruct" the signal. This is evidenced by a sharp increase in Spectral Entropy and a decrease in the Dominant Energy Ratio. The network actively disperses energy across a broader spectrum to explore and capture these challenging features, showcasing the flexibility afforded by its depth.
>
> 3. Integration and Reconstruction (Layers 9-11): In the final layers, the model's task shifts from exploration to integration. It "reconstructs" a final, efficient representation from the features learned in the middle layers. This is marked by a dramatic decrease in both Spectral Entropy and Spectral Centroid, alongside a sharp increase in the Dominant Energy Ratio to a final value of 93.8%. The network converges to a "clean", low-frequency, and highly structured representation that is optimal for the final linear layer to map to the target output.
>
> | Layer        | Spectral Centroid | Spectral Sparsity | Spectral Entropy | Dominant Energy Ratio (Top-5) |
> |--------------|-------------------|-------------------|------------------|-------------------------------|
> | FAN Layer 1  | 4.1213            | 3.4767            | 1.2264           | 0.9612                        |
> | FAN Layer 2  | 2.8760            | 5.0003            | 3.2549           | 0.7602                        |
> | FAN Layer 3  | 2.8804            | 5.0626            | 3.1556           | 0.7807                        |
> | FAN Layer 4  | 2.8810            | 4.7149            | 2.6616           | 0.8426                        |
> | FAN Layer 5  | 3.0820            | 4.5832            | 2.2248           | 0.8753                        |
> | FAN Layer 6  | 3.0815            | 5.2388            | 2.5560           | 0.8378                        |
> | FAN Layer 7  | 2.6955            | 5.8367            | 3.0115           | 0.7806                        |
> | FAN Layer 8  | 2.9132            | 5.5387            | 2.7301           | 0.8086                        |
> | FAN Layer 9  | 2.7376            | 4.1371            | 1.6760           | 0.8986                        |
> | FAN Layer 10 | 2.1266            | 3.1509            | 1.0673           | 0.9356                        |
> | FAN Layer 11 | 1.7721            | 2.9775            | 0.9270           | 0.9375                        |
> ||
>
>
>
> > Do FAN’s cosine/sine blocks hamper tasks that are intrinsically non-periodic? A negative result would be informative, and in my opinion, it would not reduce the value of the contribution.
>
> FAN's cosine/sine blocks do not hamper non-periodic tasks. Specifically, non-periodic tasks can be categorized into three types: 1) Completely non-periodic tasks: Tasks with no periodic patterns at all. 2) Implicitly periodic tasks: Tasks that are mathematically non-periodic but can be viewed as periodic from a different perspective. For example, the operation y = a + b can be seen as the result of periodic operations of single-digit addition and carrying. 3) Partially periodic tasks: Tasks where only some components are periodic, such as a rising spiral. For implicitly and partially periodic tasks, which are common in the real world, FAN can achieve good results. For completely non-periodic tasks, FAN degenerates to a standard MLP, and we have not yet identified specific scenarios where it performs negatively. We leave this for our future work.
>
> > Section on prior work says: "Previous studies have explored Fourier-based networks, but these networks generally perform well on specific tasks, while their performance on more general tasks tends to be poorer" and lists a number of citations. It would be informative to expand that sentence and section further (perhaps in the appendix) and detail more precisely how the proposed approach differs from the prior work.
>
> Thanks for your suggestions! We have expanded on this sentence as follows: Zuo and Cai (2005) proposed a control scheme based on Fourier Neural Networks, specifically for solving the tracking control problem for a class of unknown nonlinear systems. Tan (2006) studied the application of Fourier Neural Networks in the problem of aircraft engine fault classification. Jiang et al. (2022) utilized Fourier series to improve the accuracy of time series forecasting tasks. Chen et al. (2022) proposed a Fourier Imager Network that can perform end-to-end phase recovery and image reconstruction for raw holograms of novel samples. Uteuliyeva et al. investigated the poor performance of Fourier Neural Networks on general tasks such as language modeling. These works are often task-specific, and it has been proven difficult to handle more general tasks. We have added the detailed introduction of each citations to the appendix of our revised manuscript.
>
> How the proposed approach differs from the prior work: Previous Fourier-based networks face challenges in scaling to deeper networks, i.e., the capacity of their deep networks to fit the Fourier coefficients is independent of the network depth, as analyzed in Section 3. The depth scalability limits their applicability to more complex, general-purpose tasks such as language modeling. Our core differences are, "we design FAN based on the following principles: 1) the capacity of FAN to represent the Fourier coefficients should be positively correlated to its depth; 2) the output of any hidden layer can be employed to model periodicity using Fourier Series through the subsequent layers." In Section 4.3, we categorized these Fourier-based networks into three main types and compared with them, the experimental results show that only our approach achieves superior performance on both periodicity modeling task and language modeling task.
>
>
> > I suggest doing some ablation experiments to better understand what are the key contributors in FAN. For instance, freezing W_p or having just sine or cosine function or having FAN layers only in part of the network.
>
> Thanks for your suggestions.  We conduct experiments on just cosine function and having FAN layers only in part of the network,  and have added the experimental results to our revised manuscript. The results show that our current approach demonstrates a clear advantage over the variants in Periodicity Modeling and Language Modeling tasks.
>
> | Periodicity Modeling       | training loss (epoch=0) | test loss (epoch=0) | training loss (epoch=100) | test loss (epoch=100) | training loss (epoch=1000) | test loss (epoch=1000) |
> |-|-|-|-|-|-|-|
> | FAN_cos                           | 39.85                   | 63.42               | **2.67**                      | 10.18                | 1.80                       | 5.26                   |
> | FAN_replace_first_1/3_part   | 39.54                   | 46.15               | 2.95                      | 6.81                 | 1.37                       | 45.44                  |
> | FAN_replace_last_1/3_part | 42.82                   | 55.46               | 21.96                     | 27.86                | 22.79                      | 30.51                  |
> | FAN                  | **39.62**                    | **61.02**               | 2.75                      | **7.43**                 | **1.05**                       | **4.15**                   |
> ||
>
> | Language Modeling | Train Loss | In-domain Test Loss | OOD Test Loss |
> |-|-|-|-|
> | FAN_cos           | 0.2419     | 0.4802       | 0.7727        |
> | FAN_replace_first_1/3_part    | 0.2693     | 0.4313              | 0.6700        |
> | FAN_replace_last_1/3_part | **0.2417**     | 0.4660              | 0.8052        |
> | FAN               | 0.2434     | **0.4094**       | **0.6077**        |
> ||

---

> > ### Comment · Reviewer_GT2m · 2025-08-06
> >
> > Thank you for your detailed response!
> >
> > Regarding "non-periodic" tasks, you mentioned that "FAN can achieve good results", it would be helpful to provide quantitive results for that.
> >
> > Thank you for adding the results with just the cosine function - they are quite compelling.
> >
> > For the final version of the manuscript, it would be interesting to see the results with frozen W_p.

---

> > > ### Author Response · Authors · 2025-08-09
> > >
> > > Dear Reviewer GT2m,
> > >
> > > Thank you for your follow-up and constructive feedback. We are glad our previous response was helpful.
> > >
> > > Regarding the points you raised, we provide the responses as follows:
> > >
> > > 1. For "non-periodic" tasks, we would like to clarify that we had provided quantitative results on such tasks in real-world scenarios, in Section 4.2 and Appendices B.2-B.5.
> > >
> > > 2. For the cosine-only experiment, thank you for your positive feedback on this ablation study and for finding the results "quite compelling".
> > >
> > > 3. For the experiment with a frozen W_p, we have now completed it as shown below and will add it to our final version. The results show that simply freezing W_p for FAN impairs both the performance in the periodicity modeling task and harms generalization in language modeling task.
> > >
> > > | Periodicity Modeling       | training loss (epoch=0) | test loss (epoch=0) | training loss (epoch=100) | test loss (epoch=100) | training loss (epoch=1000) | test loss (epoch=1000) |
> > > |-|-|-|-|-|-|-|
> > > | freezing W_p for FAN                          | 40.52                  | **60.20**               | 15.57                      | 89.09                | 1.13                       |  156.25                |
> > > | FAN                  | **39.62**                    | 61.02               | **2.75**                      | **7.43**                 | **1.05**                       | **4.15**                   |
> > > ||
> > >
> > > | Language Modeling | Train Loss | In-domain Test Loss | OOD Test Loss |
> > > |-|-|-|-|
> > > | freezing W_p for FAN           | **0.2376**     | 0.4736      | 0.6324         |
> > > | FAN               | 0.2434     | **0.4094**       | **0.6077**        |
> > > ||
> > >
> > > We sincerely appreciate your valuable time and feedback to improve our paper.
> > >
> > > Best regards,
> > > The Authors

---

### Official Review · Reviewer_nmeh · 2025-07-03

**Clarity:** 2
**Significance:** 2
**Originality:** 2
**Rating:** 4
**Confidence:** 5

**Summary:**

The paper proposes a contribution to the well-studied fields of periodicity in neural networks and deep learning with Fourier series.

On the outset, the authors claim to introduce Fourier Analysis Networks, a purportedly novel neural network designed to effectively address periodicity modeling challenges. They claim that FANs overcomes the limitations of existing Fourier-based networks by enabling scaling to deeper and large-scale models, while simultaneously maintaining general-purpose modeling capabilities. The authors assert that FAN demonstrates superior performance in periodicity modeling tasks, especially in out-of-domain (OOD) scenarios, outperforming existing neural architectures.

However, as is detailed below in this review, the work still misses important references to previous work in the field, namely existing methods, benchmarks, metrics, and dataset introduced for this purpose, and fails to perform many comparisons with these related works. Moreover, some experimental set-ups and resulting figures and findings already appear in previous work.

In sum, I would consider the work in principle passable as a contribution of some novelty if properly embedded in the context of previous work. The singular focus on FANs as an architecture with no equal as currently alluded to in the abstract, Sections 1, 4.2, 4.3, and 5 needs to be mitigated by the appropriate acknowledgement of the existing work in the field.

**Questions:**

See weaknesses.

**Ethical Concerns:**

["NO or VERY MINOR ethics concerns only"]

**Final Justification:**

I have reviewed the author response and found it deflective. However, given the enthusiasm of other reviewers and the authors themselves, I have decided to raise the score.

It should, however, be pointed out to the authors that I did not find the claims made in the paper nor their responses to my and other reviews candid.

**Limitations:**

The authors have not done any explicit discussion of the limitations of their work. The discussion of a potential negative societal impact has not been given, but would also not be entirely necessary, given the nature of the work.

**Paper Formatting Concerns:**

No formatting concerns.

**Quality:**

2

**Strengths And Weaknesses:**

1. The work happily cites work as old as Newton 1687, yet fails to properly compare -- theoretically, experimentally, or both -- to contemporary works many of which have proposed their own methods for the same purpose:

[1] Adrian Silvescu. Fourier neural networks. In IJCNN’99. International Joint Conference on Neural Networks. Proceedings (Cat. No.
99CH36339), volume 1, pages 488–491. IEEE, 1999.

[2] Shuang Liu. Fourier neural network for machine learning. In 2013 International Conference on Machine Learning and Cybernetics, volume 1, pages 285–290. IEEE, 2013

[3] Giambattista Parascandolo, Heikki Huttunen, and Tuomas Virtanen. Taming the waves: sine as activation function in deep neural networks. 2016.

[4] Abylay Zhumekenov, Malika Uteuliyeva, Olzhas Kabdolov, Rustem Takhanov, Zhenisbek Assylbekov, and Alejandro J Castro. Fourier neural networks: A comparative study. arXiv preprint arXiv:1902.03011, 2019.

[5] Jiyoung Lee, Wonjae Kim, Daehoon Gwak, and Edward Choi. Conditional generation of periodic signals with fourier-based decoder. arXiv preprint arXiv:2110.12365, 2021.

[6] Peter Belcak and Roger Wattenhofer. Periodic extrapolative generalisation in neural networks. In 2022 IEEE Symposium Series on Computational Intelligence (SSCI), pages 1066-1073. IEEE, 2022.

2. Closely related variants of the FAN proposed in Section 3 have been also proposed in each of [1]-[5], yet comparisons in Table 1 are made only against the traditional MLP.

3. Figure 3 closely follows the exposition Figures 1&2 of [6].

4. On lines 125-130, the point of comparison should not have been MLP but each of the networks proposed in [1]-[5] across the past 25 years of activity in this research field.

5. Experimentation of Section 4.1, Appendix C, and Figure 4 mirrors that of [5] and [6] and does not acknowledge the existence of essentially equivalent synthetic benchmark in [6], which is by the virtue of the close similarity prior art. Note similar omission under point 3.

6. Experimentation of Section 4.1 glosses over RNNs, arguably the most potent architecture for the task at hand. In later experiments, LSTM is mentioned with other RNNs disregarded.

8. Table 4 acknowledges the existence of [1] and FTNNs but ignores [2]-[5] . Note similar omissions in points 2 & 4.

9. The inadequacy of snake activations described on lines 255-258 and in Appendix D is presented as a new finding but has already been pointed out and experimentally detailed in [6]. Note similar omissions in points 3 & 5.

10. The authors do not answer truthfully in the Reviewer Checklist under point 2. No explicit discussion of the limitations has been done in the paper, as their response to the checklist question indicates.

I strongly suggest authors revisit and expand the related work section, make further acknowledgement to prior art in the abstract, Section 1, Section 3, Section 4, and introduce additional baselines where possible.

---

> ### Author Rebuttal · Authors · 2025-07-31
>
> Thank you for your thorough review of the related literature and detailed feedback. We will respond to your comment point-by-point below (we merged some similar comments).
>
> > Q1: The work happily cites work ... yet fails to properly compare -- theoretically, experimentally, or both -- to contemporary works many of which have proposed their own methods for the same purpose.
>
> Thanks for your comment. We had cited works [1-5] and compared with them in our original submission, and we are grateful for your pointing out reference [6], which we have added the discussion of [6] in our revised manuscript. Our work is largely different from this work [1-6], specifically:
>
> **For work [1-4], they face challenges in scaling to deeper networks**, i.e., the capacity of their deep networks to fit the Fourier coefficients is independent of the network depth, as analyzed in Section 3. The depth scalability limits their applicability to more complex, general-purpose tasks such as language modeling. Our core differences are, "we design FAN based on the following principles: 1) the capacity of FAN to represent the Fourier coefficients should be positively correlated to its depth; 2) the output of any hidden layer can be employed to model periodicity using Fourier Series through the subsequent layers." We introduced and categorized them as Fourier Neural Networks (FNNs) [1-4] in Related Work Section. In Section 4.3, we conduct experiments to compare our approach with FNNs (we will detailed discuss the difference between FAN and FNNs in Q2), and FNNs cannot fit language modeling tasks but our approach works well.
>
> **For work [5,6], they focus on different purposes from our work** (we will detailed discuss in Q3).  And work [5] assumes all input signals have the period of 1 (as stated in page 3 of work [5]), which we conducted experiments on the same setting in Appendix B.6, and it cannot fit our periodicity modeling tasks. Work [6] is a comparative study without proposing new method, and RNN performs well in the setting of work [6], but also fails in our periodicity modeling tasks (detailed results are shown in Q6).
>
> Sorry for the misunderstanding. To enhance the clarity, we have enhanced the connection between them and added this discussion to our revised manuscript.
>
> > Q2&Q4: Closely related variants of the FAN proposed in Section 3... On lines 125-130, the point of comparison should not have been MLP...
>
> In our paper, we compare our approach with not only MLP but also previous Fourier-based networks in experiments of Section 4.3. The reason why we formally analyze the differences with MLP in Section 3 and lines 125-130 is that our goal is to explore the likelihood of FAN to offer broad applicability like MLP.
>
> We provide the analysis of work [1-4] compared to our approach in below table and have added it to the appendix of our revised version. We mainly discuss work [1-3], due to the work [4] is a comparative study without proposing new method. The work [2, 3] can be regraded as the simplified version of work [1], where work [2,3] are fully equivalent to work [1] when hyperparameter m=1 (They can be converted to each other through trigonometric identities, and the differences in their parameters can be absorbed by the learnable W and B). The work [5] cannot fit our periodicity modeling tasks, so we didn't compared with it.
>
> | Metric  | FAN Layer    | Layer of [1]                               | Layer of [2]    | Layer of [3]                     |
> |-|-|-|-|-|
> | **Formula**    | $[\cos(W_p x) \Vert \sin(W_p x) \Vert \sigma(B_{\bar{p}} + W_{\bar{p}} x)]$ | $W_f \Pi_m \cos(W_{a_m} x+B_{a_m})+B_f$   | $W_{f_c} \cos(W_{a_c} x+B_{a_c}) + B_{f_c} + W_{f_s} \sin(W_{a_s} x+B_{a_s}) + B_{f_s}$ | $W_f \sin(W_a x+B_a)+B_f$        |
> | **Num Params** | $(1-\frac{d_p}{d_o}) (d_i d_o + d_o)$                                    | $m(d_i d_h + d_h) + d_o d_h + d_o$        | $d_i (d_c + d_s) + (d_c + d_s) + d_o (d_c + d_s) + 2d_o$                                | $d_i d_h + d_h + d_o d_h + d_o$  |
> | **FLOPs**      | $(1-\frac{d_p}{d_o}) \times 2d_i d_o + \gamma d_o$                       | $2m d_h d_i + d_h(m-1) + 2d_h d_o + \gamma m d_h$ | $2d_i (d_c + d_s) + 2d_o (d_c + d_s) + \gamma (d_c + d_s)$                              | $2d_h(d_i + d_o) + \gamma d_h$   |
>
> ### Notation Key:
> - $d_i = d_{\text{input}}$ (Input dimension hyperparameter)
> - $d_o = d_{\text{output}}$ (Output dimension hyperparameter)
> - $d_p$ = FAN layer hyperparameter (default $\frac{1}{4}d_o$)
> - $d_h$ = Hidden dimension hyperparameter (for [1], [3])
> - $d_c, d_s$ = Layer [2] hyperparameters (cosine/sine branch dimensions)
> - $m$ = Layer [1] hyperparameter (number of projections)
> - $\gamma$ = FLOPs per nonlinear activation ($\sigma$, $\cos$, or $\sin$)
>
> > Q3 & Q5: Figure 3 closely follows the exposition Figures 1&2 of [6]. Experimentation of Section 4.1, Appendix C, and Figure 4 mirrors...
>
> Our work and work [6] focus on different problems, with only the display form looking like similar. Specifically, the model's input and output in Figure 3 are both scalars, i.e., function input and function output, and we show the results of entire test dataset. The input of the network in Figures 1 & 2 of [6] is the sequence of previous step, and the output is the value of the next step, and they show the median model prediction and the shaded regions after 30 runs of test sample.  This distinction explains their use of RNN. While an RNN excels in the settings of [6], it performs poorly on our periodicity modeling task (See results in Q6).
>
> Moreover, experiments in Section 4.1, Appendix D (Appendix C in Q5 is maybe a typo. Appendix C includes the experimental setup of real-world tasks, which is not related to work [5,6]), and Figure 4 are also different. First, Figure 4 analyzes the entire model training process from another perspective, proving that its inability to fit periodicity is not due to underfitting. There is no similar experiment in work [5,6]. Second, Section 4.1 and Appendix D includes the experiments of periodicity modeling task, which are different from the problems studied in work [5,6].
>
> > Q6: Experimentation of Section 4.1 glosses over RNNs...
>
> We compared with LSTM, considering it is generally considered to be a powerful and effective variant in the RNN family. To address your concerns, we have supplemented the results of the standard RNN and another variant of RNN, GRU, below. The results show that our approach outperforms both RNN and GRU on various tasks, including periodicity modeling (the  experiment in Section 4.1), time series torecasting, and language modeling.
>
> | Periodicity Modeling       | training loss (epoch=0) | test loss (epoch=0) | training loss (epoch=100) | test loss (epoch=100) | training loss (epoch=1000) | test loss (epoch=1000) |
> |-|-|-|-|-|-|-|
> | RNN          | 44.44                    | 49.12                | 32.60                      | 37.52                  | 24.71                       | 190.58                  |
> | GRU          | 44.69                    | 39.76                | 35.95                      | 37.76                  | 20.58                       | 40.00                   |
> | Ours          | **39.62**                   | **61.02**                | **2.75**                       | **7.43**                  | **1.05**                        | **4.15**                    |
> ||
>
> | Time Series Forecasting | MSE | MAE |
> |-|-|-|
> | RNN          | 1.369                   | 0.829             |
> | GRU          | 0.986                   | 0.694                |
> | Ours          | **0.839**                   | **0.631**                |
> ||
>
> | Language Modeling | Train Loss | In-domain Test Loss | OOD Test Loss |
> |-|-|-|-|
> | RNN   | 0.5036     | 0.5264       | 0.6829        |
> | GRU   | 0.4811     | 0.5049       | 0.6392        |
> | Ours   | **0.2434**     | **0.4094**       | **0.6077**  |
> ||
>
> > Q7: Table 4 acknowledges the existence of [1] and FTNNs but ignores [2]-[5]...
>
> We would like to clarify that we did not ignore work [2-5] in Table 4, where we compared the performance of our approach FAN, FNN, FSNN, FTNN, and MLP on language modeling tasks. Please note that FNN and FSNN are more capable generalizations of [1-4] and [5], respectively. As mentioned in Q2, work [2-4] can be regarded as a simplified version of FNN [1]. And work [5] can be regarded as a simplified version of FSNN, where it assumes all input signals have the period of 1. Therefore, in our comparison with FNN [1] and FSNN, we are, in fact, considering all the works [1-5].
>
> > Q8: The inadequacy of snake activations described on lines 255-258 ...
>
> The comparison with snake in the appendix is not proposed as a new finding, but only illustrates its shortcomings. We have modified the statement and added the discussion of work [6] in our revised manuscript.
>
> > Q9: The authors do not answer truthfully in the Reviewer Checklist under point 2..
>
> Thanks for your feedback. Due to space limitations in the main text, we only discussed the limitations of our work in the Justification of Checklist. We have added a section on limitations to our revised manuscript:
>
> First, we only demonstrate the effectiveness of FAN on some mainstream real-world tasks (including symbolic formula representation, time series forecasting, language modeling, image recognition, etc.), and we aim to further broaden the applicability of FAN in our future work. Second, although we have explored the generalizability of FAN and confirmed that FAN outperforms the baseline method in some real-world tasks, the boundaries of this model's generalizability remain unknown. However, we have not yet identified specific scenarios where it performs poorly. We leave this for our future work.

---

> ### Author Response · Authors · 2025-08-04
> **Looking forward to your feedback**
>
> Dear Reviewer nmeh,
>
> Do our responses address your concerns? Please let us know if there are any other questions you'd like to discuss! We are happy to address them while there is still time. Thanks again for your time and detailed review!
>
> Best regards,
> Authors

---

> > ### Author Response · Authors · 2025-08-06
> >
> > Dear Reviewer nmeh,
> >
> > Thank you again for your time and for your detailed review of our paper.
> >
> > As the discussion period is coming to a close soon, we just wanted to gently check if our rebuttal has addressed the concerns you raised in your review.
> >
> > Please feel free to let us know if there are any other questions you'd like to discuss. Your feedback is very important to us, and we are on standby to provide any further clarification.
> >
> > Best regards,
> > The Authors

---

> > > ### Comment · Reviewer_nmeh · 2025-08-06
> > >
> > > I have read the author response and the reviews of other reviewers. While I still consider the embedding of the paper in the existing literature somewhat unfaithful to the state of this field of research, I have come to the conclusion that the community would nevertheless benefit from the inclusion of this works in the conference proceedings. I have raised my score.

---

> > > > ### Author Response · Authors · 2025-08-07
> > > >
> > > > Dear Reviewer nmeh,
> > > >
> > > > We are very grateful for your support and for your decision to raise your score. Thank you for recognizing that the community would benefit from the inclusion of this work in the conference proceedings.
> > > >
> > > > For properly embedding our paper in the existing literature, we have added reference [6] to the related work section of our revised version with a detailed discussion, as you suggested. For the other references, [1-5], while they were cited in our initial draft, we agree that our previous discussion was insufficient. Therefore, we have taken this opportunity to significantly strengthen our comparison with these works to address your concerns.
> > > >
> > > > We believe these revisions have made the related work section of our paper much more robust. Thank you for your guidance, and we appreciate your time and effort you provided throughout this process. We are happy to discuss any further questions at any time.
> > > >
> > > > Best regards,
> > > > Authors

---

### Official Review · Reviewer_kgu3 · 2025-07-03

**Clarity:** 4
**Significance:** 3
**Originality:** 2
**Rating:** 5
**Confidence:** 3

**Summary:**

This paper introduces the Fourier Analysis Networks (FAN), which extends the usual MLP networks to model periodicity in inputs by adding sine and cosine as activation functions to each layer. For motivation, this paper notes that sine and cosine together generate the Fourier Basis in Fourier Series expansion (Section 3).

With the added sine and cosine activation functions in each layer, this paper argues that periodicity modeling is available throughout the network, and together with MLP activations in each layer this periodicity modeling capacity scales with network depth (Sections 1 and 3).

For validation, this paper conducts experiments with synthetic data (Section 4.1, Periodicity Modeling) and real-world data (Section 4.2, Symbolic Formula Representation, Time Series Forecasting, Language Modeling), showing that FAN (or FAN Gated) gets lower loss (\~7%, MSE or MAE) and higher accuracy (\~5%), using fewer params (\~15%) and less FLOPs, than competing (non-periodicity-aware) models (such as MLP, KAN, Transformers without FAN, LSTM, Mamba).

Even when comparing with other periodicity-aware, Fourier-based models (FNN, FSNN, FTNN), the newly introduced Fourier Analysis Networks (FAN) still performs better in synthetic data (in Periodicity Modeling, Section 4.3) with lower loss and fewer params.

**Questions:**

Currently in a FAN Layer $\phi(x) = [\cos(W_px) || \sin(W_px) || \sigma(B_{\bar p}+W_{\bar px})]$ (Figure 2), so cosine and sine do not have the bias components ($B_p$), unlike the MLP part. Would adding the bias component help, for example, in modeling “phase shift” in periodicity? Note that after adding bias components, the network can use sine or cosine alone, without using both. In particular, how would such networks flair in experiments?

**Ethical Concerns:**

["NO or VERY MINOR ethics concerns only"]

**Limitations:**

### Note on periodic generalization

Lines 78-79, Page 3: “The power of Fourier Series lies in its ability to represent a wide variety of functions, including non-periodic functions through periodic extensions, enabling the extraction of frequency components.”

Such periodic extensions assume that the generalization is periodic (outside of the primary period), which may or may not hold.

The examples in Figure 1, Figure 3 and Figure 6 (See Section 4.1) are the best possible periodic generalization for Fourier Analytic Networks: outside the training set, the test set is perfect periodic extension. So by “over-fitting” to the training set, we get perfect generalization to the test set, which is what FAN does best, compared with other networks (MLP, KAN, Transformers).

But to be fair, the experimental results in Section 4.2 show that periodicity are indeed present in real-world data, e.g., in Time Series, or in Language Modeling.

**Paper Formatting Concerns:**

No.

**Quality:**

4

**Strengths And Weaknesses:**

For strengths, the experiments shows that FAN gets lower loss with fewer params in real-world data such as time series forecasting and language modeling, and the added sine and cosine activation functions to model periodicity are well motivated, as elaborated in Section 3.

For weaknesses, the periodicity ratio ($d_p$) is yet another hyper-parameter to tune.

---

> ### Author Rebuttal · Authors · 2025-07-31
>
> Thank you for your in-depth review and valuable feedback. We will respond to your comment point-by-point below.
>
> > Currently in a FAN Layer (Figure 2), so cosine and sine do not have the bias components (B_p), unlike the MLP part. Would adding the bias component help, for example, in modeling “phase shift” in periodicity? Note that after adding bias components, the network can use sine or cosine alone, without using both. In particular, how would such networks flair in experiments?
>
> Thanks for your suggestions. We conduct experiments on the setting of adding bias components and use cosine alone, namely FAN_cos, and have added the experimental results to our revised manuscript.  Overall, FAN outperforms FAN_cos on periodicity modeling and language modeling tasks. Although FAN_cos occasionally achieves lower training loss, its significantly higher test loss indicates poor generalization.
>
> | Periodicity Modeling       | training loss (epoch=0) | test loss (epoch=0) | training loss (epoch=100) | test loss (epoch=100) | training loss (epoch=1000) | test loss (epoch=1000) |
> |-|-|-|-|-|-|-|
> | FAN_cos              | 39.85                    | 63.42               | **2.67**                      | 10.18                | 1.80                       | 5.26                   |
> | FAN                  | **39.62**                    | **61.02**               | 2.75                      | **7.43**                 | **1.05**                       | **4.15**                   |
> ||
>
> | Language Modeling | Train Loss | In-domain Test Loss | OOD Test Loss |
> |-|-|-|-|
> | FAN_cos           | **0.2419**     | 0.4802       | 0.7727        |
> | FAN               | 0.2434     | **0.4094**       | **0.6077**        |
> ||
>
> > periodicity ratio (d_p) is yet another hyper-parameter to tune.
>
> We fix the periodicity ratio to 1/4 and achieve stable and clear improvements across all experiments with this same setting. Although we found in the appendix that further adjusting the periodicity ratio can lead to even greater improvements for specific task, we believe that the default parameters used in this paper can already cover common scenarios.
>
> > Note on periodic generalization ...
>
> We do believe that some generalization properties can be expressed as periodicity, so we aim to extract the information with periodic generalization using the Fourier Principle, which leads to FAN naturally performing better on tasks with periodicity. Of course, for real-world data, periodic extension is a valid and parsimonious assumption that complies with Occam's razor. However, the extent of its generalization capability is likely highly dependent on the specific real-world task. We have added this discussion to our revised manuscript.

---

> > ### Comment · Reviewer_kgu3 · 2025-08-02
> >
> > > Thanks for your suggestions. We conduct experiments on the setting of adding bias components and use cosine alone, namely FAN_cos, and have added the experimental results to our revised manuscript. Overall, FAN outperforms FAN_cos on periodicity modeling and language modeling tasks. Although FAN_cos occasionally achieves lower training loss, its significantly higher test loss indicates poor generalization.
> >
> > Thanks for following up with this suggestion of handling phase shifts. Regarding poorer generalizations of cosine-only networks:
> >
> > 1. For Periodicity Modeling, are the data perfectly periodic (in terms of $\sin$ and $\cos$) with no phase shift? ($y = \sin(x)$ in Figure 1, $y = \sin(x + \sin(2x))$ in Figure 4, etc., instead of, e.g., $y = \sin(x + 0.5)$ or $y = \sin(x + 0.5 + \sin(2x))$? If so, it is not unreasonable to expect $\sin$ and $\cos$ performs better, because the data are generated to suit them. What about performance on shifted periodic data?
> >
> > 2. For Language Modeling, it is good to validate that $\sin$ and $\cos$ together, performs better than cosine-only plus phase-shift. Although it remains to explain this finding, given that after learning the coefficients, the Fourier expansion explanation (where $\cos$ and $\sin$ receive perfect multiples of roots of unity) does not necessarily hold, so intuitively they have no advantage over cosine-only plus phase-shift.

---

> ### Author Response · Authors · 2025-08-04
> **Response to Reviewer kgu3**
>
> Thanks for your comments. In the last rebuttal response, the Periodicity Modeling experiments was conducted on a complex compound periodic function $y = exp(\sin(\pi x)^2 + \cos(x) + x \mod 3 - 1$. To address your question about performance on shifted periodic data, we ran two new experiments: 1) $y = \sin(z + 0.5)$; 2) $y = \sin(w + 0.5) + \sin(2w)$.  The results show that even on shifted periodic data, the generalization performance of FAN on the test data is much better than FAN_cos.
>
> | Periodicity Modeling (Task)       | training loss (epoch=0) | test loss (epoch=0) | training loss (epoch=100) | test loss (epoch=100) | training loss (epoch=1000) | test loss (epoch=1000) |
>  |-|-|-|-|-|-|-|
>  | $y = \sin(z + 0.5)$              |                         |                     |                           |                      |                            |                        |
>  | FAN_cos                           | **0.2922**                  | **0.6014**              | 7.66e-04                  | 2.35e-02             | 3.71e-04                   | 1.97e-04               |
>  | FAN                               | 0.4194                  | 0.8037              | **3.75e-04**                  | **4.49e-03**             | **3.09e-04**                   | **3.13e-05**               |
>  | $y = \sin(w + 0.5) + \sin(2w)$   |                         |                     |                           |                      |                            |                        |
>  | FAN_cos                           | **0.5552**                  | **0.6303**             | 2.41e-03                  | 5.30e-02             | 1.74e-03                   | 1.85e-03               |
>  | FAN                               | 0.5782                  | 0.6835              | **1.95e-03**                  | **3.09e-02**             | **1.08e-03**                   | **2.48e-04**               |
> ||
>
> Moreover, we would like to provide a straightforward explanation for the finding in language modeling from the following perspective: Imagine that cos(Wx) and sin(Wx) are two fundamental "periodic building blocks", like red and blue LEGO bricks.
> 1. Periodic Part of FAN Layer ($[\cos(W_p x) || \sin(W_p x)]$) gives you both the red brick (cos) and the blue brick (sin) separately. The next layer of the network can then freely decide how much red and how much blue to use, allowing it to construct any combination of phase (the shade of color) and amplitude (the size of the brick). This is extremely flexible.
> 2. Periodic Part of  Cosine-Only+Phase Shift ($[\cos(W_p x+B_p)]$) is like being given a pre-mixed "purple" brick, where the red and blue components have already been combined in a fixed ratio. The next layer can only decide whether to use this purple brick and with how much strength (adjusting its amplitude), but it cannot change the inherent "color mixture" (the phase) of the brick itself.
>
> The core advantage of FAN is that the FAN layer gives the network greater freedom. By providing independent cos and sin building blocks, the network can more easily and precisely learn any periodic function it requires, much like a painter who has all the primary colors can mix any hue they desire. The "cosine-only + phase shift" method restricts this creative potential, making the learning process more difficult. This disadvantage becomes even more pronounced when stacking multiple layers to construct complex functions.
>
> We hope our responses can address your problems. Thanks again for your dedicated time and insightful feedback.

---

> > ### Comment · Reviewer_kgu3 · 2025-08-07
> >
> > Thank you for following up with experiments and detailed responses.
> >
> > While it is hard to argue against the experimental results, however I am still curious about the theoretical understanding, especially regarding using both sine-and-cosine-without-phase-shift vs cosine(or sine)-only-with-phase-shift.
> >
> > The main idea is that “$\cos$ with phase shift can express $\sin$” (and conversely “$\sin$ with phase shift can express $\cos$”, namely, $\sin(x) = \cos(x - \pi/2)$ (and $\cos(x) = \sin(x + \pi/2)$). Hence with phase shifts ($B_p$ which contains $\pm\pi/2$), a $\cos$-only block (or a $\sin$-only block) can simulate a $\cos$-and-$\sin$ block (especially without phase-shifts).
> >
> > I think it might be hard for the neural network to learn the shifts $\pm\pi/2$, or it is beneficial to just learn how to use $\cos$ and $\sin$ in the next layer (as you suggested). For otherwise, a $\cos$-only-with-phase-shift block is more flexible in choosing the ratio to mix (by controlling how many $\pm\pi/2$ in $B_p$), than a $\cos$-and-$\sin$ block (which only allow the same number of $\cos$ and $\sin$).

---

> > > ### Author Response · Authors · 2025-08-07
> > >
> > > Dear Reviewer kgu3,
> > >
> > > Thank you very much for your continued follow-up and profound theoretical insights!  In fact, the "cosine-only with bias" approach you mentioned was one of our initial designs. However, through extensive experiments, we found that its real-world task performance and generalization ability is weaker than our current architecture of FAN.
> > >
> > > We completely agree with your assessment, and we believe the two hypotheses you proposed get to the heart of the matter:
> > >
> > > 1) It is hard for a neural network to precisely learn and fix to a specific phase shift like π/2.  This would require the network to learn a large number of paired, highly-structured cosine units: their weight would need to be identical, while the corresponding elements in their bias would need to differ by exactly ±π/2. The network can easily get stuck in local optima and fail to discover this relationship.
> > >
> > > 2) It is more effective to directly provide sin and cos as orthogonal basis functions (i.e., building blocks) to the subsequent layer than having the network expend effort on "learning how to create a sine function from a cosine". This allows the network to efficiently learn how to combine these components to represent any phase and amplitude.
> > >
> > > Thank you again for your valuable questions. This discussion has significantly helped us articulate why FAN performs better than FAN_cos with greater clarity. We sincerely appreciate the time and effort you have dedicated to reviewing our paper.
> > >
> > > Best regards,
> > > Authors

---

> > > > ### Comment · Reviewer_kgu3 · 2025-08-09
> > > >
> > > > Thanks you for the reply.
> > > >
> > > > I don’t have much to add, besides saying that I learned a lot reading the paper and in the discussion.

---

> > > > > ### Author Response · Authors · 2025-08-09
> > > > >
> > > > > Dear Reviewer kgu3,
> > > > >
> > > > > Thank you for your positive feedback. This has been a valuable learning experience for us as well, thanks to your insightful comments, and we are truly grateful for your dedication to our manuscript.
> > > > >
> > > > > Best regards,
> > > > > Authors

---

### Note · Authors · 2025-08-16

Dear reviewers and chairs,

We would like to express our sincere gratitude to all reviewers for their positive and insightful feedback. We are grateful for the reviewers' acknowledgement of our work as a “technically solid paper, with high impact” (Reviewers kgu3 and GT2m), its “well-explained motivation, valuable contribution, and convincing results” (Reviewer rvp8), and especially for the supportive comment that "the community would nevertheless benefit from the inclusion of this work in the conference proceedings" (Reviewer nmeh).

We have addressed the concerns raised by the reviewers, and have incorporated the clarifications and revisions in our response into the final version of the paper. For the major concern raised by Reviewer nmeh, i.e., properly embedding our paper in the existing literature, we have added reference [6] to Related Work section of our revised manuscript with a detailed discussion, as Reviewer nmeh suggested. For the other references, [1-5], while they were cited in our initial draft, we have additionally provided a deeper analysis of them compared with our approach, as described in response, and added the discussion to our revised manuscript.

Our work introduces an effective and scalable method for modeling periodicity in deep neural networks, a fundamental challenge in many domains, and we believe it is a valuable contribution to the field. We sincerely appreciate the time and effort invested by all the reviewers and chairs.

Best regards,
Authors

---

### Decision · Program_Chairs · 2025-09-17

**Decision:**

Accept (poster)

**Comment:**

This paper develops a novel neural network that effectively addresses the challenges of periodicity modeling. All reviews are positive, specifically: A, BA, A, BA. Reviewers kgu3 and GT2m commend the work as "a technically solid paper with high impact"; Reviewer rvp8 recognizes it for its "well-explained motivation, valuable contribution, and convincing results." Most concerns have been adequately resolved. Therefore, I recommend the acceptance of this submission. Additionally, I expect that the authors will incorporate the suggested modifications from the rebuttal phase into the final version.